# Learning to Refine: Spectral-Decoupled Iterative Refinement Framework for Precipitation Nowcasting

Yunlong Zhou [1 2]   Chen Zhao [1 2]   Danyang Peng [1 2]   Fanfan Ji [1 2]   Xiao-Tong Yuan [1 2]

## Abstract

Accurate precipitation nowcasting is vital for disaster mitigation, but deep learning methods face a key trade-off: regression models produce over-smoothed, spectrally decaying predictions that blur convective details and violate turbulence power laws; diffusion models generate realistic yet unanchored hallucinations lacking physical grounding. We propose Spectral-Decoupled Iterative Refinement (SDIR), a deterministic framework that reformulates nowcasting as progressive frequency-decoupled refinement. SDIR first extracts a stable low-frequency synoptic skeleton, then iteratively refines high-frequency textures under physical constraints, eliminating both blurring and hallucinations. It features a dual-path design: the Synoptic Frequency-Guided Former (SFG-Former) with Scale-Adaptive Transformers for global structure, and the Fourier Residual Refiner (FR-Refiner) with Scale-Conditioned Fourier Neural Operators for fine residuals. A Physically Consistent Power Spectral Density (PCPSD) loss with dynamic masking enforces a turbulence-consistent spectral distribution. Experiments on three benchmarks show SDIR significantly outperforms SOTA methods in spatial accuracy while achieving spectral fidelity competitive with diffusion-based methods, enabling reliable high-resolution operational nowcasting. Code link: https://github.com/RuntimeWarning/SDIR.

## 1. Introduction

Precise precipitation nowcasting, predicting high-resolution rainfall intensity within a short-term horizon (0–2 hours),

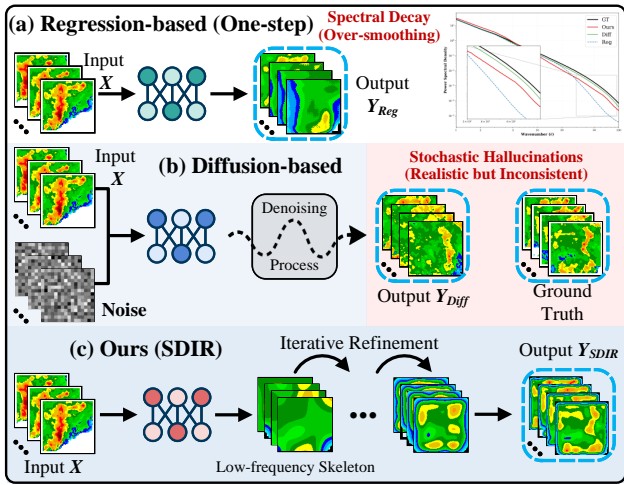

*Figure 1.* Paradigm comparison in precipitation nowcasting. (a) Regression: Suffers from spectral decay and loss of peak intensity, falling significantly below the ground truth (GT) in the PSD plot. (b) Diffusion: Generates realistic high-frequency details but lacks physical grounding, resulting in stochastic hallucinations inconsistent with the GT. (c) SDIR (Ours): Reformulates nowcasting as deterministic spectral evolution, progressively restoring high-frequency details from a low-frequency synoptic skeleton.

is a cornerstone of modern meteorological services (Shi et al., 2015). Its accuracy is paramount for urban flood management, aviation safety, and disaster mitigation (Pan et al., 2024b). However, as climate volatility intensifies, the demand for forecasts that are both physically consistent and spatially fine-grained has reached an unprecedented level (Pan et al., 2024a).

Early radar-based extrapolation methods, such as optical flow (Ayzel et al., 2019), effectively capture short-term motion but struggle with non-linear convective growth and decay. The recent paradigm shift toward deep learning has significantly advanced the field (Pan et al., 2025). Initial spatiotemporal models based on CNNs and RNNs, particularly ConvLSTM (Shi et al., 2015), greatly improved feature extraction by integrating convolutional operations into recurrent structures. More recently, generative frameworks—including Generative Adversarial Networks (GANs) (Goodfellow et al., 2020) and diffusion models—have pushed boundaries further, generating high-

[1]School of Intelligence Science and Technology, Nanjing University, Suzhou, 215163, China. [2]State Key Laboratory for Novel Software Technology, Nanjing University, Nanjing, 210023, China. Correspondence to: Xiao-Tong Yuan <xtyuan@nju.edu.cn>.

*Proceedings of the 43rd International Conference on Machine Learning*, Seoul, South Korea. PMLR 306, 2026. Copyright 2026 by the author(s).

fidelity precipitation fields that maintain structural clarity over extended lead times (Ho et al., 2020; Ravuri et al., 2021; Yu et al., 2024).

Despite these advancements, a fundamental trade-off persists, as illustrated in Figure 1. Traditional spatial-domain regression models (Figure 1(a)) suffer from a well-known over-smoothing effect (Mathieu et al., 2016; Zhou et al., 2026). Driven by pixel-wise objectives like MSE, these models minimize loss by averaging potential outcomes under spatial uncertainty, inevitably eroding sharp convective details into blurred patterns. This regression-to-the-mean behavior not only suppresses peak rainfall intensities but also disrupts the natural energy cascade across spatial scales, resulting in physically inconsistent forecasts that deviate from turbulence power laws (Kolmogorov, 1991; Skamarock, 2004). In contrast, diffusion models (Figure 1(b)) (Gao et al., 2023) attempt to recover these details via stochastic sampling from noise, producing visually realistic convective cells. However, this stochasticity often leads to "unanchored hallucinations"—generating structures that appear plausible but lack grounding in the input data's physical context, such as misplaced rain cells or exaggerated intensities.

This trade-off highlights why nowcasting requires a refinement-based approach: precipitation evolution is inherently multi-scale and progressive, starting with stable large-scale motions that serve as boundary conditions for smaller-scale details (Houze Jr, 2018). A single-pass model struggles to balance global stability with local sharpness without either blurring (regression) or hallucinating (stochastic generation). To address this, we need a deterministic framework that decouples the prediction into frequency bands, iteratively refining from coarse low-frequency skeletons to fine high-frequency textures (Yu et al., 2025; Zhao et al., 2026). This ensures physical anchoring at every stage, mitigating hallucinations while countering over-smoothing by explicitly supervising spectral fidelity.

We propose the Spectral-Decoupled Iterative Refinement (SDIR) framework, which reformulates precipitation nowcasting as a physically grounded, coarse-to-fine spectral refinement process, as shown in Figure 1(c). Unlike diffusion models that begin from stochastic noise and risk physically inconsistent predictions, SDIR deterministically extracts a stable low-frequency skeleton from the historical sequence, which serves as a reliable anchor for subsequent refinements. It then uses this skeleton along with historical data as dual conditions to progressively unlock high-frequency textural details through multi-step, frequency-guided inference. The frequency decoupling is crucial because it mirrors atmospheric physics: low frequencies capture global synoptic patterns (e.g., storm fronts) that must be stabilized first to constrain high-frequency local textures (e.g., convective cells), ensuring the entire spectrum adheres to turbulence

laws. This is realized via a two-stage hybrid architecture: the Synoptic Frequency-Guided Former (SFG-Former) employs Scale-Adaptive Transformers to capture long-range spatiotemporal dependencies and produce a sharp, physically consistent low-frequency skeleton as a robust foundation. The Fourier Residual Refiner (FR-Refiner) leverages Scale-Conditioned Fourier Neural Operators (SFNO) to synthesize high-frequency residual textures, compensating for spectral decay in a fully deterministic manner. To enforce physical consistency, we introduce a Physically Consistent Power Spectral Density (PCPSD) loss with dynamic masking, which ensures adherence to atmospheric turbulence power laws throughout the refinement process. By bridging the gap between stochastic generative detail and deterministic reliability, SDIR transforms nowcasting into a robust, physically plausible refinement process.

The core contributions of this work are outlined below.

- We propose the SDIR framework, which reformulates precipitation nowcasting as a multi-stage deterministic spectral evolution process, effectively bridging the trade-off between generative realism and structural reliability by iteratively refining from anchored low-frequency skeletons to high-frequency details.

- We introduce a hybrid dual-path architecture that synergizes the global structural modeling of the SFG-Former with the local high-frequency synthesis of the FR-Refiner, achieving seamless integration of long-range meteorological dependencies and spectral fidelity.

- We present a PCPSD loss equipped with a dynamic masking mechanism that enforces adherence to atmospheric turbulence power laws by aligning the model's energy distribution with the ground truth spectrum.

- Extensive experiments on three public benchmarks demonstrate that SDIR effectively mitigates blurring in longer-lead-time forecasts, outperforming state-of-the-art (SOTA) methods in spatial accuracy while achieving competitive or superior spectral realism.

## 2. Related Work

### 2.1. Regression-based Models

The field of precipitation nowcasting was significantly advanced by ConvLSTM (Shi et al., 2015), which integrated convolutions into recurrent neural networks to capture spatiotemporal correlations. Subsequent works, such as TrajGRU (Shi et al., 2017), PredRNN (Wang et al., 2017), PredRNN++ (Wang et al., 2018), and MIM (Wang et al., 2019), improved modeling of non-linear motion and non-stationary dynamics through trajectory offsets, memory

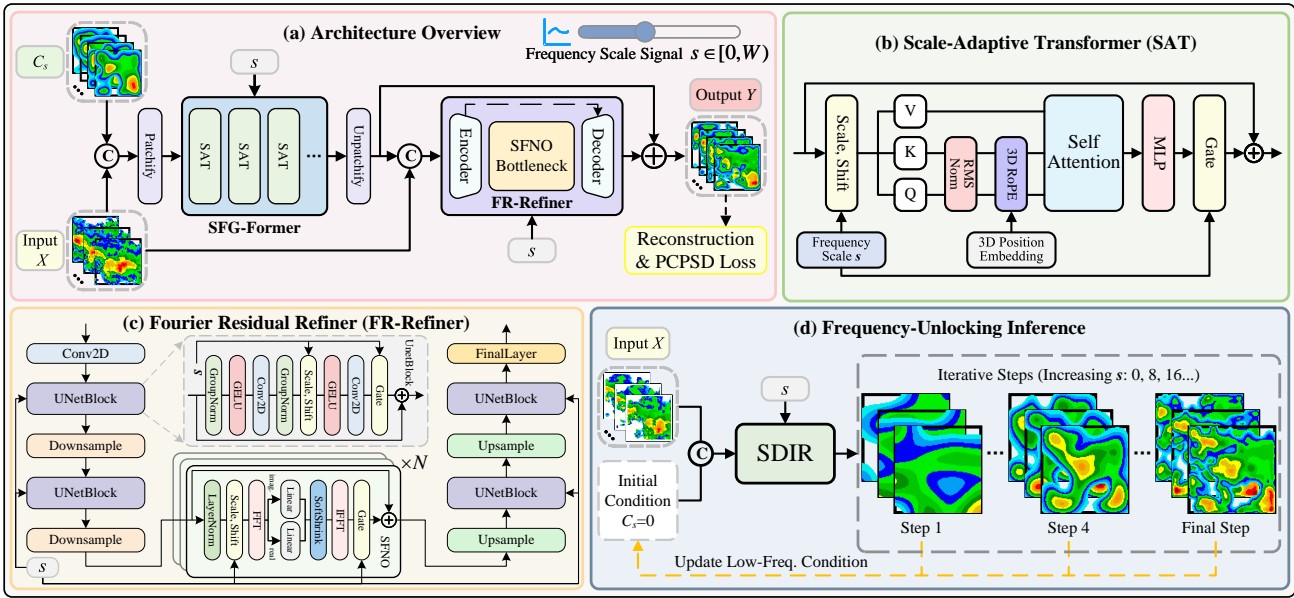

*Figure 2.* The overall architecture and operational paradigm of SDIR. (a) Architecture Overview: SDIR couples the SFG-Former for synoptic skeleton extraction with the FR-Refiner for spectral detail synthesis. The model is optimized using a combination of reconstruction and PCPSD losses. (b) Scale-Adaptive Transformer. (c) Fourier Residual Refiner. (d) Frequency-Unlocking Inference: During inference, SDIR starts from a zero state ($s = 0$) and performs iterative steps with increasing $s$, updating the low-frequency condition $C_s$ to progressively synthesize high-resolution details.

flows, and differential units. Physics-informed approaches like PhyDNet (Guen & Thome, 2020) introduced deep disentanglement to separate dynamics from residuals. With the rise of self-attention, Transformer-based models have set new benchmarks: Earthformer (Gao et al., 2022a) proposed cuboid attention for large-scale modeling, Rainformer (Bai et al., 2022) focused on multi-scale patterns, and SwinL-STM (Tang et al., 2023) enhanced long-range dependencies. More recently, AlphaPre (Lin et al., 2025) decoupled radar amplitude and phase for separate motion and intensity modeling. Despite these advances in deterministic performance, most methods overlook multi-scale energy distribution, resulting in the well-documented spectral decay that compromises high-frequency details and physical realism.

### 2.2. Generation-based Models

To recover high-frequency details, generative approaches have gained increasing attention. Early GAN-based methods, such as MPL-GAN (Liu & Lee, 2020) and STEAN (Zhou et al., 2024), showed that adversarial training can produce sharper radar echoes. DGMR (Ravuri et al., 2021) marked a major advance with a conditional GAN featuring a large latent generator and dedicated spatiotemporal discriminators for statistical fidelity. NowcastNet (Zhang et al., 2023) proposed a hybrid physics-conditional framework combining deterministic evolution with generative refinement to better capture extreme events. Recent diffusion-based models have shifted the paradigm toward iterative

denoising (Zhao et al., 2024): PreDiff (Gao et al., 2023) adapted latent diffusion for enhanced visual realism, Cas-Cast (Gong et al., 2024) introduced cascaded diffusion for multi-scale feature resolution, and DiffCast (Yu et al., 2024) proposed a residual diffusion strategy where a deterministic backbone predicts the main trend and diffusion handles stochastic residuals for better frequency alignment. However, these approaches face a persistent trade-off: stochastic hallucinations, where visually plausible convective cells appear in spatially inconsistent locations or violate physical constraints, ultimately limiting their operational reliability.

## 3. Methodology

### 3.1. Overall Framework and Physical Intuition

To reconcile the inherent trade-off between structural reliability and fine-grained spectral realism, we propose the SDIR framework. Our design is inspired by the energy cascade in atmospheric dynamics (Kolmogorov, 1991), where kinetic energy is transferred from large-scale synoptic systems to small-scale turbulent eddies. As illustrated in Figure 2, SDIR operationalizes this hierarchy by decoupling the forecasting task into two specialized, frequency-aware stages within an end-to-end unified network.

**Stage I: Synoptic Foundation (SFG-Former).** This stage focuses on the "where"—capturing the deterministic motion and large-scale structural evolution. By processing the historical sequence $X$ alongside a low-frequency prior $C_s$,

it yields $\hat{Y}_{base}$, representing the stable synoptic skeleton.

**Stage II: Convective Refinement (FR-Refiner).** This stage addresses the how "intense"—recovering high-frequency textures suppressed during global modeling. Utilizing a Fourier-enhanced architecture, it synthesizes the high-frequency residual $\hat{Y}_{res}$ to enrich the base skeleton.

During inference, SDIR performs deterministic multi-step refinement. By iteratively increasing the frequency scale signal $s$, the model progressively "unlocks" higher-frequency bands, transforming a blurred structural backbone into a high-resolution, physically consistent forecast.

### 3.2. Spectral-Aware Training Curriculum

The core of SDIR's capability lies in its spectral curriculum, which mimics the nested nature of weather systems (Hu et al., 2026). We introduce a frequency scale signal $s \in \{0, 1, \ldots, W - 1\}$, where $W$ is the spatial resolution. This signal acts as a "spectral regulator" that controls the bandwidth of the low-frequency condition $C_s$.

For each training sample $Y$, the low-frequency condition $C_s$ is constructed via 2D Discrete Cosine Transform (DCT) truncation:

$$C_s = \text{IDCT}\left(\text{Trunc}_{s \times s}\left(\text{DCT}(Y)\right)\right), \qquad (1)$$

where $\text{Trunc}_{s \times s}(\cdot)$ retains only the top-left $s \times s$ (lowest-frequency) coefficients, acting as an ideal low-pass filter.

To ensure the model masters coarse structures before tackling complex textures, we sample $s$ from a non-uniform Beta distribution, biasing the training focus toward the foundational synoptic scales:

$$s = \lfloor W \cdot \sigma \rfloor, \quad \sigma \sim \text{Beta}(\alpha = 1.0, \beta = 3.0). \quad (2)$$

### 3.3. Synoptic Frequency-Guided Former (SFG-Former)

Serving as the foundational branch of our dual-path architecture, the SFG-Former is designed to capture large-scale synoptic evolution and provide a stable structural backbone for rainfall fields. At its core, it leverages SAT blocks (Labs et al., 2025) to ensure consistent conditioning on the prescribed spectral depth. To preserve the intrinsic spatiotemporal structure of weather systems, we incorporate 3D Rotary Position Embeddings (3D RoPE) (Su et al., 2024). This enables fully translation-invariant modeling across both spatial and temporal dimensions (Hu et al., 2025).

The historical sequence $X \in \mathbb{R}^{B \times T_{\text{in}} \times C \times H \times W}$ and the low-frequency condition $C_s \in \mathbb{R}^{B \times T_{\text{out}} \times C \times H \times W}$ are first concatenated along the temporal dimension. The fused sequence is then partitioned into non-overlapping 2D patches of size $p \times p$ (spatially) and projected into a latent embedding $z \in \mathbb{R}^{B \times L \times D}$.

In each SAT block, a Frequency Scale Embedder (FSE) maps the scalar $s$ into a modulation triplet $(\gamma, \beta, \alpha)$:

$$\gamma, \beta, \alpha = \mathbf{W}_{\text{mod}}\left(\text{SiLU}(\text{FSE}(s))\right) + \mathbf{b}_{\text{mod}}. \qquad (3)$$

The latent tensor $z$ is normalized and modulated to obtain the spectral-aware hidden state:

$$z_{\text{mod}} = (1 + \gamma) \odot \text{LayerNorm}(z) + \beta. \qquad (4)$$

The block output is then computed via a gated residual connection:

$$z_{\text{out}} = z + \alpha \odot \text{Transformer}(z_{\text{mod}}), \qquad (5)$$

where $\text{Transformer}(\cdot)$ denotes the standard multi-head self-attention followed by a feed-forward network. This frequency-aware modulation ensures the skeleton is robust across scales, providing a strong anchor for refinement.

### 3.4. Fourier Residual Refiner (FR-Refiner)

The FR-Refiner is a frequency-aware, U-Net-like residual generator designed to recover the high-frequency details and spectral fidelity lost in the patch-based SFG-Former. As illustrated in Figure 2(c), the refiner takes the concatenation of the historical sequence $X$ and the initial structural prediction $\hat{Y}_{base}$ as input, conditioned on the frequency scale signal $s$ to ensure alignment with the target spectral depth:

$$\hat{Y}_{res} = \text{FR-Refiner}([X \| \hat{Y}_{base}], s). \qquad (6)$$

The backbone adopts a hierarchical encoder-decoder structure (Tian et al., 2025), utilizing PixelUnshuffle and PixelShuffle for resolution transitions to preserve fine-grained spatial details. Residual skip connections bridge multi-scale encoder features and the decoder, effectively fusing low-level textures with high-level semantics.

To explicitly model long-range frequency interactions, we place a series of SFNO blocks (Guibas et al., 2021) at the latent bottleneck. Unlike conventional spatial convolutions, which are local and limited in capturing cross-scale couplings, SFNO performs fully global operations directly in the frequency domain via Fourier transforms. Within each SFNO block, input features are first projected into the spectral domain via a 2D Fast Fourier Transform (FFT). The resulting complex signal is then decomposed into its real and imaginary components, which undergo efficient interactions through multi-layer linear transformations. The complex coefficients are processed by a SoftShrink operator and subsequently reconstructed into the spatial domain via an Inverse FFT (IFFT).

To ensure the refinement remains responsive to the evolving spectral depth, the scale signal $s$ is processed by a series of FSEs and injected into each block via Adaptive Normalization (AN). Finally, an AdaLN-modulated output layer

aggregates these refined features, yielding a high-precision residual map that complements the synoptic foundation. The ultimate precipitation forecast $\hat{Y}$ is achieved through the additive fusion of the global synoptic foundation and the synthesized high-frequency residual:

$$\hat{Y} = \hat{Y}_{base} + \hat{Y}_{res}. \tag{7}$$

### 3.5. Physically Consistent Power Spectral Density Loss

Pixel-wise losses like MSE promote over-smoothing by averaging errors, neglecting spectral distributions essential for physical realism. To preserve multi-scale structures, mitigate spectral decay, and adhere to atmospheric turbulence laws (Kolmogorov, 1991), we propose the PCPSD loss. It supervises frequency-domain statistics by matching radially averaged PSD, aligning energy distribution across scales with turbulence theory.

**Radial PSD Estimation.** Given a prediction $\hat{Y}$ and ground truth $Y$, we first apply a 2D Hann window $w$ to reduce edge artifacts. The 2D power spectrum is then estimated using the real FFT (rFFT):

$$P(k_y, k_x) = \left| \mathrm{rFFT}(Y \odot w) \right|^2 / (HW), \tag{8}$$

where $\odot$ denotes element-wise multiplication, and $H, W$ are spatial dimensions. To compensate for the energy discarded in the one-sided rFFT spectrum, we double the power of all positive frequencies except the Nyquist frequency. The 1D isotropic radial PSD $S(k)$ is obtained by averaging over annular bins:

$$S(k) = \frac{1}{|B_k|} \sum_{(k_y, k_x) \in B_k} P(k_y, k_x), \tag{9}$$

where $B_k$ is the set of frequency coordinates in the $k$-th radial bin, and $k \in [0, 1]$ denotes the normalized radial wavenumber (typically represented by the bin center).

**Dynamic Frequency Masking and Weighting.** A core innovation of SDIR is the Dynamic Frequency Masking mechanism, tightly aligned with the frequency-unlocking curriculum. The loss is computed in the log-spectral domain to emphasize relative energy differences across scales:

$$\mathcal{L}_{\mathrm{pcpsd}} = \frac{\sum_k \Omega(k, s) \cdot (\log S_{\mathrm{pred}}(k) - \log S_{\mathrm{gt}}(k))^2}{\sum_k \Omega(k, s)}, \tag{10}$$

where $\Omega(k, s)$ is a dynamic weighting function controlled by the scale signal $s$.

Specifically,

$$\Omega(k, s) = (k + \epsilon)^\gamma \cdot \begin{cases} 0.2 & \text{if } k \le k_s(s) \\ 1.0 & \text{if } k > k_s(s), \end{cases} \tag{11}$$

with $\gamma$ as the boost exponent, and $k_s(s) = s/W$ as the normalized cutoff wavenumber unlocked at the current scale $s$. This scheme applies a mild global boost to higher wavenumbers to counteract natural spectral decay, while assigning full weight to the currently unlocked frequency bands for strong supervision and reduced weight to lower frequencies already stabilized by the synoptic skeleton.

### 3.6. Multi-Task Learning Objective

We employ a dual-path training strategy to synergistically reconcile spatial accuracy with physical consistency. The total objective $\mathcal{L}$ combines spatial and spectral constraints:

$$\mathcal{L} = \mathcal{L}_{\mathrm{base}} + \mathcal{L}_{\mathrm{res}} + \phi(s)\mathcal{L}_{\mathrm{pcpsd}}, \tag{12}$$

where $\mathcal{L}_{\mathrm{base}}$ and $\mathcal{L}_{\mathrm{res}}$ denote the $L_1$ spatial losses for the synoptic foundation and residual map, respectively. We introduce a dynamic schedule $\phi(s) = \eta \cdot (s/W)^2$ that progressively increases the penalty on spectral deviations as the refinement granularity increases. Here, the base coefficient $\eta = 0.01$ balances the PCPSD loss against the spatial loss, preventing the spectral term from dominating the training objective while retaining its regularizing effect.

### 3.7. End-to-End Training

Algorithm 1 outlines the spectrally decoupled training curriculum. In each iteration, a frequency scale signal $s$ is sampled from a Beta distribution skewed toward low frequencies, which controls the spectral depth of the low-frequency condition $C_s$ via DCT truncation. This non-uniform sampling ensures the model first masters coarse synoptic structures before progressively tackling finer convective textures, mirroring the hierarchical nature of atmospheric dynamics. Under this curriculum, the SFG-Former learns to produce a stable structural skeleton from increasingly restricted low-frequency priors, while the FR-Refiner is trained to synthesize the complementary high-frequency residuals conditioned on this skeleton. The two branches are jointly optimized under a multi-task objective that combines spatial

---

**Algorithm 1** SDIR Training Process
___

**Input:** Sequence $X$, Ground truth $Y$, Resolution $W$.
**Parameters:** $\theta_1$ (SFG-Former), $\theta_2$ (FR-Refiner).
**repeat**
    Sample $\sigma \sim \mathrm{Beta}(\alpha, \beta)$, set $s = \lfloor W \cdot \sigma \rfloor$.
    $C_s = \mathrm{IDCT}(\mathrm{Trunc}_{s \times s}(\mathrm{DCT}(Y)))$.
    $\hat{Y}_{base} = \mathrm{SFG\text{-}Former}(X, C_s, s; \theta_1)$.
    $\hat{Y}_{res} = \mathrm{FR\text{-}Refiner}([X \| \hat{Y}_{base}], s; \theta_2)$.
    $\hat{Y} = \hat{Y}_{base} + \hat{Y}_{res}$.
    $\mathcal{L} = \mathcal{L}_{base} + \mathcal{L}_{res} + \phi(s)\mathcal{L}_{pcpsd}$.
    Update $\theta_1, \theta_2$ via AdamW.
**until** Convergence
___

**Algorithm 2** SDIR Inference Process

**Input:** Sequence $X$, Schedule $\mathcal{S} = \{s_1, \ldots, s_K\}$, $s_1 = 0$.

**Initialize:** $C_s = 0$.

**for** $k = 1$ **to** $K$ **do**

    $s = \mathcal{S}[k]$

    $\hat{Y}_{base}^{(k)} = \text{SFG-Former}(X, C_s, s)$.

    $\hat{Y}_{res}^{(k)} = \text{FR-Refiner}([X \| \hat{Y}_{base}^{(k)}], s)$.

    $\hat{Y}^{(k)} = \hat{Y}_{base}^{(k)} + \hat{Y}_{res}^{(k)}$.

    **if** $k < K$ **then**

        $C_s = \text{IDCT}(\text{Trunc}_{s_{k+1} \times s_{k+1}}(\text{DCT}(\hat{Y}^{(k)})))$.

    **end if**

**end for**

**Return:** $\hat{Y}^{(K)}$.

$L_1$ losses with the PCPSD spectral loss, where the dynamic schedule $\phi(s)$ progressively amplifies the spectral penalty as $s$ increases, enforcing turbulence-consistent energy distributions at finer scales. This co-training strategy ensures that both spatial accuracy and physical spectral consistency are achieved in a unified end-to-end framework.

### 3.8. Frequency-Unlocking Inference

During inference, SDIR follows a "coarse-to-fine unlocking" schedule, as described in Algorithm 2. Starting from a "cold start" condition ($s = 0$), the model iteratively processes the historical inputs and the previous prediction, incrementally expanding the frequency bandwidth according to a predefined schedule $\mathcal{S} = s_1, s_2, \ldots, s_K$. At each step $k$, the current prediction $\hat{Y}^{(k-1)}$ is re-truncated to scale $s_k$ and reused as a structural conditioning signal for the next iteration. This progressive refinement ensures that large-scale synoptic structures are stabilized before the model synthesizes complex local textures, effectively suppressing meteorological hallucinations and promoting the stable evolution of high-intensity convective cells.

## 4. Experiments

### 4.1. Datasets

We evaluate our approach on three widely adopted precipitation nowcasting benchmarks: CIKM (101×101 resolution), Shanghai (501×501 resolution), and SEVIR (384×384 resolution). These datasets cover diverse geographical regions, providing a comprehensive testbed for assessing model performance across varying spatial scales and weather patterns.

**CIKM.** This dataset contains 14,000 radar echo sequences, each consisting of 15 consecutive frames captured at 6-minute intervals (Luo et al., 2021). We follow a standard split of 8,000 sequences for training, 2,000 for validation,

and 4,000 for testing.

**Shanghai.** The Shanghai dataset comprises composite reflectivity products from a dual-polarization WSR-88D radar located in Pudong, Shanghai, consisting of 1,534 training and 526 testing sequences (Chen et al., 2020; Yu et al., 2024). Each sequence contains 25 consecutive frames at a 6-minute temporal resolution.

**SEVIR.** This is a large-scale multimodal meteorological benchmark (Veillette et al., 2020). We specifically utilize the NEXRAD radar mosaics of Vertically Integrated Liquid (VIL), consisting of 47,624 sequences for training, 12,080 for validation, and 16,212 for testing. Each sequence comprises 25 frames recorded at 5-minute intervals.

### 4.2. Evaluation Metrics

To comprehensively evaluate the SDIR framework, we adopt a suite of metrics that assess deterministic reliability, physical consistency, and perceptual quality across operational nowcasting scenarios. Categorical accuracy is measured using the Critical Success Index (CSI) and Heidke Skill Score (HSS) across multiple reflectivity thresholds, capturing the model's performance at varying precipitation intensities. Pixel-wise intensity deviations are quantified with the Mean Absolute Error (MAE). Additionally, the Structural Similarity Index (SSIM) (Wang et al., 2004) is employed to evaluate the morphological and structural fidelity of the generated precipitation fields.

### 4.3. Implementation Details

All experiments are conducted on four NVIDIA RTX 4090D GPUs using the AdamW optimizer with an initial learning rate of 0.0003. The SDIR framework integrates eight SAT blocks (hidden dimension 512) in the SFG-Former and eight SFNO blocks at the FR-Refiner's latent bottleneck. Following the architectural design of FourCastNet (Pathak et al., 2022), the SFNO block retains all frequency components and applies a fixed soft-thresholding scalar of 0.01 to sparsify them adaptively. The FR-Refiner adopts a U-Net-like structure with two U-Net blocks per down/upsampling stage and channel dimension doubling from a base of 32. To balance computational efficiency and hardware constraints, input resolutions are standardized to 128×128 for CIKM (via zero-padding) and 256×256 for Shanghai and SEVIR (via bilinear downsampling). For the CIKM and Shanghai datasets, performance is evaluated at intensity thresholds of 20, 30, 35, and 40 dBZ, while for the SEVIR dataset, the thresholds are 16, 74, 133, 160, 181, and 219.

### 4.4. Compared With Different Models

In this section, we compare our proposed SDIR framework against eight SOTA methods: ConvLSTM (Shi et al., 2015),

*Table 1.* Quantitative comparison of the proposed SDIR framework against SOTA baselines on the CIKM dataset.

| Models | HSS ↑ | | | | | CSI ↑ | | | | | SSIM ↑ | MAE ↓ |
|---|---|---|---|---|---|---|---|---|---|---|---|---|
| | 20 | 30 | 35 | 40 | AVG | 20 | 30 | 35 | 40 | AVG | | |
| ConvLSTM | 0.5198 | 0.3201 | 0.2220 | 0.1949 | 0.3142 | 0.5258 | 0.2438 | 0.1502 | 0.1262 | 0.2615 | 0.4860 | 738.05 |
| PredRNN | 0.6172 | 0.3707 | 0.2654 | 0.2414 | 0.3737 | 0.6465 | 0.3291 | 0.2033 | 0.1648 | 0.3359 | 0.5157 | 784.84 |
| PhyDNet | 0.6197 | 0.4118 | 0.3432 | 0.2763 | 0.4128 | 0.6599 | 0.3392 | 0.2418 | 0.1842 | 0.3563 | 0.4306 | 694.99 |
| SimVP | 0.6060 | 0.3839 | 0.2680 | 0.1593 | 0.3543 | 0.6407 | 0.3059 | 0.1777 | 0.0944 | 0.3047 | 0.4482 | 707.43 |
| Earthformer | 0.6035 | 0.4234 | 0.3461 | 0.2906 | 0.4159 | 0.6363 | 0.3408 | 0.2447 | 0.1956 | 0.3544 | 0.4903 | 674.99 |
| MIMO | 0.6253 | 0.3842 | 0.3006 | 0.2675 | 0.3944 | 0.6680 | 0.3270 | 0.2118 | 0.1725 | 0.3448 | 0.5180 | 687.67 |
| DiffCast | 0.6126 | 0.4080 | 0.3364 | 0.2714 | 0.4071 | 0.6443 | 0.3296 | 0.2363 | 0.1806 | 0.3477 | 0.4710 | 669.01 |
| AlphaPre | 0.6058 | 0.3643 | 0.2732 | 0.2098 | 0.3633 | 0.6272 | 0.2813 | 0.1881 | 0.1402 | 0.3092 | 0.4775 | 661.40 |
| **Ours** | **0.6558** | **0.4807** | **0.4089** | **0.3440** | **0.4724** | **0.6885** | **0.3971** | **0.2958** | **0.2358** | **0.4043** | **0.5574** | **600.37** |

*Table 2.* Quantitative comparison of the proposed SDIR framework against SOTA baselines on the Shanghai dataset.

| Models | HSS ↑ | | | | | CSI ↑ | | | | | SSIM ↑ | MAE ↓ |
|---|---|---|---|---|---|---|---|---|---|---|---|---|
| | 20 | 30 | 35 | 40 | AVG | 20 | 30 | 35 | 40 | AVG | | |
| ConvLSTM | 0.5464 | 0.4156 | 0.3007 | 0.1780 | 0.3602 | 0.4260 | 0.2999 | 0.2052 | 0.1134 | 0.2611 | 0.7438 | 1846.2 |
| PredRNN | 0.6223 | 0.5122 | 0.3934 | 0.2396 | 0.4419 | 0.4934 | 0.3777 | 0.2727 | 0.1556 | 0.3248 | 0.8156 | 1487.8 |
| PhyDNet | 0.6530 | 0.5796 | 0.4913 | 0.3574 | 0.5203 | 0.5255 | 0.4393 | 0.3522 | 0.2399 | 0.3892 | 0.8133 | 1386.0 |
| SimVP | 0.5776 | 0.4408 | 0.3308 | 0.1758 | 0.3812 | 0.4339 | 0.2964 | 0.2056 | 0.0990 | 0.2587 | 0.7747 | 1572.5 |
| Earthformer | 0.6546 | 0.5679 | 0.4722 | 0.3112 | 0.5015 | 0.5230 | 0.4249 | 0.3332 | 0.2034 | 0.3711 | 0.7643 | 1395.8 |
| MIMO | 0.6275 | 0.5104 | 0.3848 | 0.1874 | 0.4275 | 0.4911 | 0.3637 | 0.2516 | 0.1101 | 0.3041 | 0.8085 | 1413.0 |
| DiffCast | 0.6350 | 0.5460 | 0.4530 | 0.3338 | 0.4920 | 0.5051 | 0.4058 | 0.3185 | 0.2216 | 0.3628 | 0.8080 | 1450.1 |
| AlphaPre | 0.6284 | 0.4979 | 0.3750 | 0.2089 | 0.4276 | 0.4943 | 0.3639 | 0.2611 | 0.1386 | 0.3145 | 0.7534 | 1445.3 |
| **Ours** | **0.7033** | **0.6379** | **0.5648** | **0.4466** | **0.5882** | **0.5779** | **0.4952** | **0.4167** | **0.3089** | **0.4497** | **0.8548** | **1129.1** |

PredRNN (Wang et al., 2017; 2022), PhyDNet (Guen & Thome, 2020), SimVP (Gao et al., 2022b), Earthformer (Gao et al., 2022a), MIMO (Ning et al., 2023), DiffCast (Yu et al., 2024), and AlphaPre (Lin et al., 2025). The quantitative results are summarized in Tables 1 to 3.

Across all experimental evaluations, the proposed SDIR framework consistently achieves state-of-the-art performance, outperforming all baselines across every metric and reflectivity threshold. On the CIKM dataset, SDIR delivers the highest average HSS and CSI, surpassing the strongest competitor, Earthformer, by 13.6% and 14.1%, respectively. Our model sustains its performance advantage on the Shanghai dataset, securing the best results in all categories, including pixel-level accuracy (MAE) and structural fidelity (SSIM). Notably, even at the most challenging 40 dBZ threshold, SDIR remains robust while traditional architectures like ConvLSTM and SimVP suffer precipitous declines. Furthermore, these uniform gains extend to the large-scale SEVIR dataset, validating the robust generalization and high precision of SDIR in modeling complex radar echo dynamics.

Figure 3 illustrates the performance decay over a 120-minute lead time on the Shanghai dataset. While all models exhibit

inherent accuracy decline and rising cumulative error as the prediction interval expands, SDIR consistently maintains the highest scores across HSS, CSI, and SSIM. Although competitors like AlphaPre and PhyDNet perform competitively in short-term forecasts (0–30 min), their performance drops more sharply as the horizon extends. The widening performance gap in the 60–120 minute range highlights the

*Table 3.* Quantitative comparison of the proposed SDIR framework against SOTA baselines on the SEVIR dataset. The HSS and CSI metrics represent the mean values across multiple intensity thresholds. Detailed performance results for specific intensity thresholds are provided in the Appendix.

| Models | HSS | CSI | SSIM | MAE |
|---|---|---|---|---|
| ConvLSTM | 0.3512 | 0.2715 | 0.6062 | 2896.9 |
| PredRNN | 0.3772 | 0.2991 | 0.5703 | 3058.9 |
| PhyDNet | 0.4172 | 0.3311 | 0.7063 | 2103.3 |
| SimVP | 0.2674 | 0.2149 | 0.6530 | 2623.5 |
| Earthformer | 0.4066 | 0.3230 | 0.6706 | 2241.8 |
| MIMO | 0.3687 | 0.2873 | 0.6970 | 2326.8 |
| DiffCast | 0.3972 | 0.3057 | 0.6690 | 2595.5 |
| AlphaPre | 0.4052 | 0.3193 | 0.6100 | 2463.0 |
| **Ours** | **0.4401** | **0.3499** | **0.7544** | **1897.9** |

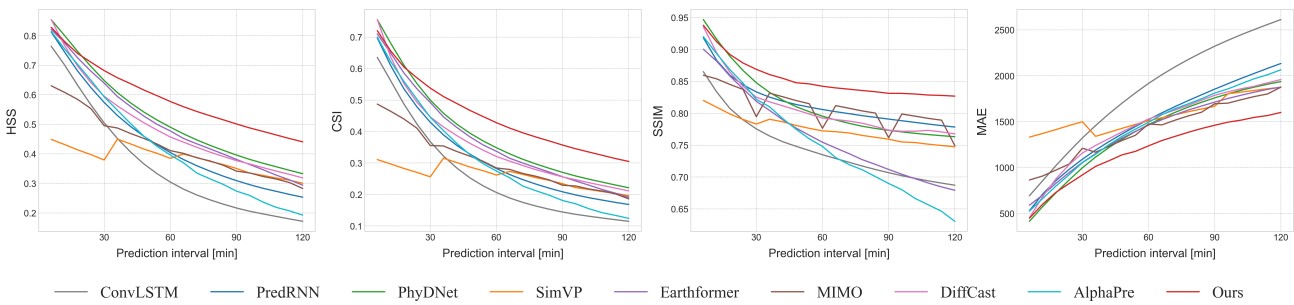

*Figure 3.* Quantitative comparison of different models on the Shanghai dataset. The plots illustrate the performance across four metrics over a 120-minute forecast horizon.

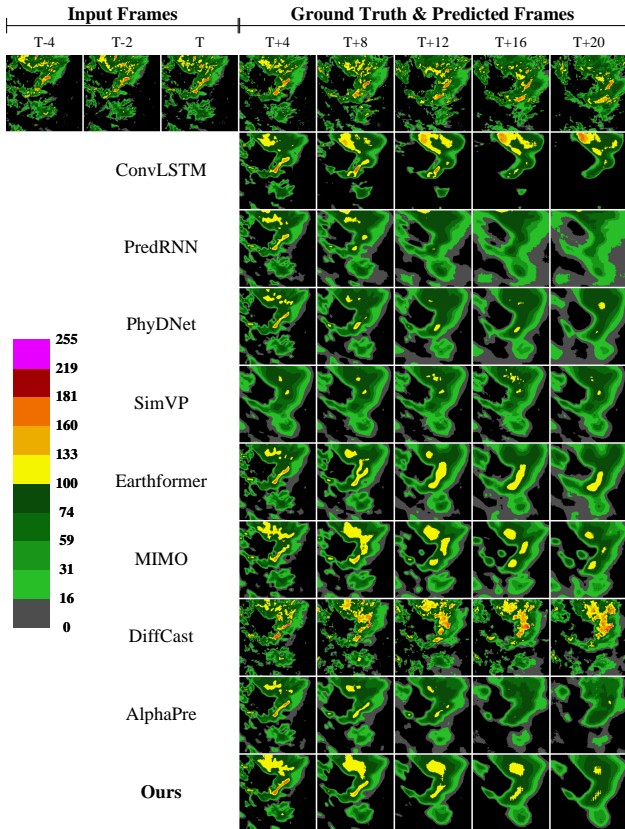

*Figure 4.* Qualitative comparison of precipitation nowcasting results on the SEVIR dataset. For additional representative cases, please refer to the Appendix.

superior ability of SDIR to capture long-range temporal dependencies and effectively mitigate error accumulation.

A qualitative comparison of the prediction results is provided in Figure 4. Visual inspection reveals that recurrent-based models, such as ConvLSTM, PredRNN, and PhyD-Net, suffer from significant blurring beyond the T+12 frame, failing to maintain the morphology of the echo cells. While Earthformer tracks general motion, it generates oversmoothed boundaries and simplified structures in high-intensity regions, leading to a loss of fine-grained

textures. In contrast, DiffCast produces fragmented and noisy artifacts that lack spatial coherence. Notably, the frames generated by SDIR are the most consistent with the ground truth; even at the maximum lead time of T+20, SDIR successfully preserves the clear boundaries and internal structures of high-reflectivity cores. This qualitative evidence confirms that our framework effectively addresses the over-smoothing problem, providing more realistic and meteorologically meaningful nowcasting results.

## 4.5. Ablation Studies

We conduct comprehensive ablation experiments on the Shanghai dataset to quantify the contributions of our architectural components, training strategies, inference step configurations, and Beta distribution parameters, with results summarized in Tables 4 to 7.

### 4.5.1. EFFECT OF MODEL COMPONENTS

Table 4 demonstrates the synergistic contributions of each component. The standalone SFG-Former (Exp. a) provides strong structural fidelity, while the standalone FR-Refiner (Exp. b) improves categorical scores but results in higher MAE. Their integration (Exp. c) yields balanced improvements across all evaluation criteria. Crucially, further incorporating the PCPSD loss (Exp. d) achieves optimal performance across all metrics. This underscores the effectiveness of the PCPSD loss in enforcing turbulence-consistent spectral fidelity while enhancing spatial precision.

### 4.5.2. EFFECT OF TRAINING CONFIGURATIONS

Table 5 examines the influence of loss functions, sampling strategies, and feature modulation. Training with pure MAE loss yields excellent pixel-wise accuracy but severely underperforms on HSS and CSI, underscoring its limitation in capturing multi-scale convective dynamics. Switching from our Beta-distributed sampling to uniform sampling causes a sharp degradation in CSI and HSS, confirming the value of prioritizing low-frequency structures as stable refinement anchors. Removing AN likewise leads to substantial drops

across metrics, highlighting its role in dynamically aligning intermediate features with the target spectral level $s$.

*Table 4.* Ablation study of SDIR components on the Shanghai dataset. S-I and S-II denote the SFG-Former and the FR-Refiner, respectively.

| Exp. | S-I | S-II | PCPSD | HSS | CSI | SSIM | MAE |
|---|---|---|---|---|---|---|---|
| (a) | ✓ | | | 0.3529 | 0.2559 | 0.8478 | 1248.8 |
| (b) | | ✓ | | 0.4614 | 0.3266 | 0.8125 | 1586.1 |
| (c) | ✓ | ✓ | | 0.5367 | 0.4057 | 0.8512 | 1138.3 |
| **Ours** | ✓ | ✓ | ✓ | **0.5882** | **0.4497** | **0.8548** | **1129.1** |

*Table 5.* Ablation study of loss functions, sampling strategies, and injection mechanisms. $\mathcal{L}_{base+res}$ denotes the entire model trained with only one MAE loss. "Uniform" denotes sampling $s \sim \mathcal{U}(0,1)$. AN refers to adaptive normalization.

| Configuration | HSS | CSI | SSIM | MAE |
|---|---|---|---|---|
| $\mathcal{L}_{base+res}$ | 0.4509 | 0.3420 | **0.8555** | **1106.4** |
| Uniform | 0.2842 | 0.2097 | 0.8458 | 1284.1 |
| w/o AN | 0.5073 | 0.3725 | 0.8102 | 1476.4 |
| **Ours** | **0.5882** | **0.4497** | 0.8548 | 1129.1 |

*Table 6.* Performance comparison across different inference steps.

| Steps | HSS | CSI | SSIM | MAE | Time |
|---|---|---|---|---|---|
| 1-step | 0.5584 | 0.4243 | 0.8522 | **1111.0** | **0.30s** |
| 8-step | 0.5882 | **0.4497** | **0.8548** | 1129.1 | 1.17s |
| 16-step | **0.5898** | 0.4487 | 0.8533 | 1190.1 | 2.14s |
| 32-step | 0.5564 | 0.4164 | 0.8475 | 1352.6 | 4.09s |

*Table 7.* Sensitivity to the Beta distribution.

| Distribution | HSS | CSI | SSIM | MAE |
|---|---|---|---|---|
| $\text{Beta}(0.8, 1.5)$ | 0.5606 | 0.4206 | 0.8485 | 1299.2 |
| $\text{Beta}(1.0, 2.5)$ | 0.4835 | 0.3759 | 0.8493 | 1241.5 |
| $\text{Beta}(1.0, 3.0)$ | **0.5882** | **0.4497** | **0.8548** | **1129.1** |
| $\text{Beta}(1.0, 3.5)$ | 0.5582 | 0.4222 | 0.8543 | 1140.5 |

4.5.3. EFFECT OF THE NUMBER OF INFERENCE STEPS

Table 6 analyzes performance across refinement steps. While 1-step prediction yields the lowest MAE, insufficient iterative refinement leads to over-smoothing, weakening its HSS and CSI. Progressive refinement gradually recovers high-frequency details, reaching an optimal trade-off at 8 steps, which achieves peak CSI and SSIM with a competitive MAE. Beyond 8 steps, excessive refinement introduces artifacts that degrade performance; while 16 steps yield a marginal improvement in HSS, this comes at the cost of increased MAE, decreased SSIM, and nearly doubled

inference time. Therefore, we select 8-step inference as our default configuration, striking the best balance between predictive accuracy and computational efficiency.

4.5.4. SENSITIVITY TO THE FREQUENCY CURRICULUM

Table 7 reports model performance under different Beta distributions used to sample the frequency scale $s$ during training. The configuration $\text{Beta}(1.0, 3.0)$ achieves the best performance across all metrics, benefiting from a moderate low-frequency emphasis that stabilizes large-scale structures before fine-scale details are learned. Increasing the skewness to $\text{Beta}(1.0, 3.5)$ degrades categorical metrics, suggesting that an overly aggressive low-frequency bias limits the recovery of fine-scale convective structures.

## 5. Conclusion

We introduced SDIR, a deterministic multi-stage spectral refinement framework that effectively resolves the fundamental trade-off between spatial over-smoothing and stochastic hallucination in deep learning-based precipitation nowcasting. By progressively refining from a stable low-frequency synoptic skeleton to high-frequency convective textures, SDIR achieves both structural reliability and realistic detail. The dual-path design synergizes the global spatiotemporal modeling of the SFG-Former with the scale-conditioned Fourier residual synthesis of the FR-Refiner, while the PCPSD loss with dynamic masking enforces turbulence-consistent spectral distributions across lead times. Experiments on multiple public benchmarks show that SDIR significantly outperforms state-of-the-art methods in spatial accuracy while delivering competitive spectral fidelity. Frequency-decoupled iterative refinement offers a robust pathway toward physically plausible, high-resolution operational nowcasting.

**Limitations and Future Work.** While SDIR excels in maintaining structural integrity and physical consistency throughout the extended forecast horizon, it exhibits noticeably lower fine-scale texture realism compared to generative models like DiffCast. Future research will explore the integration of more sophisticated texture-synthesis modules to bridge this visual fidelity gap while preserving the deterministic reliability of the synoptic field.

## Acknowledgements

This work was supported by the Natural Science Foundation of China (No. U21B2049); the Special Fund for Key Program of Science and Technology of Jiangsu Province (No. BG2024042); the Gusu Leading Talents Program for Innovation and Entrepreneurship (No. ZXL2025323); the Fundamental and Interdisciplinary Disciplines Breakthrough Plan of the Ministry of Education of China under Grant (No.

JYB2025XDXM118); and the "111 Center" (No. B26023). We thank the ICML reviewers for their valuable feedback and suggestions.

## Impact Statement

This paper presents a spectral-decoupled iterative refinement method for precipitation nowcasting that aims to produce sharper, better-aligned forecasts, potentially improving short-term severe weather. The proposed method is computationally efficient at inference, requiring only 8 iterative steps on standard GPU hardware, making it practical for operational deployment in real-world meteorological services.

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

## A. More Details about Datasets

**CIKM** The Conference on Information and Knowledge Management (CIKM) dataset[1] originates from the CIKM AnalytiCup 2017 competition and consists of weather radar observations collected in Shenzhen, Guangdong Province, China. Each radar echo map has a spatial resolution of $101 \times 101$ pixels, with each pixel corresponding to a 1 km $\times$ 1 km ground area. Pixel values are linearly converted to radar reflectivity $Z$ (dBZ) using the following transformation:

$$Z = \text{pixel\_value} \times \frac{95}{255} - 10. \tag{13}$$

The dataset provides observations at four vertical levels—0.5 km, 1.5 km, 2.5 km, and 3.5 km above ground level—with 1 km spacing. Following common practice in previous works, we use the 3.5 km altitude scans for both training and evaluation.

**Shanghai** The dataset spans from August 2015 to July 2018, comprising 127,447 radar images captured at approximately 6-minute intervals. To facilitate model training, the raw polar-coordinate data were interpolated onto a $0.01° \times 0.01°$ longitude-latitude grid. For numerical stability, the logarithmic reflectivity $Z$ is normalized to the range $[0, 1]$ via $\hat{Z} = Z/70$.

**SEVIR** The Storm Event Imagery Dataset (SEVIR) is a large-scale, spatiotemporally aligned multimodal benchmark for meteorology. The dataset comprises approximately 10,000 independent weather events, each containing a 49-frame sequence with 5-minute time increments. SEVIR provides a comprehensive suite of weather observations across five distinct modalities: three channels from the GOES-16 geostationary satellite, NEXRAD weather radars, and the GOES-16 Geostationary Lightning Mapper flashes.

## B. Detailed Definitions of Evaluation Metrics

To provide a rigorous quantitative assessment, we detail the calculation of categorical metrics used in this study. All scores are derived from a pixel-wise confusion matrix calculated at specific reflectivity thresholds. The elements of the matrix are defined as:

- **True Positive (TP):** Pixels correctly predicted as exceeding the threshold (hits).

- **True Negative (TN):** Pixels correctly predicted as staying below the threshold (correct negatives).

- **False Positive (FP):** Pixels incorrectly predicted as exceeding the threshold (false alarms).

- **False Negative (FN):** Pixels exceeding the threshold but predicted to be below (misses).

**Heidke Skill Score (HSS)** The HSS measures the forecast accuracy relative to a random chance forecast. Unlike simple accuracy, it accounts for the correct predictions of both precipitation and non-precipitation events, offering a balanced evaluation even when precipitation pixels are sparse. It is defined as:

$$\text{HSS} = \frac{2 \times (\text{TP} \times \text{TN} - \text{FP} \times \text{FN})}{(\text{TP} + \text{FP})(\text{FP} + \text{TN}) + (\text{TP} + \text{FN})(\text{FN} + \text{TN})}. \tag{14}$$

**Critical Success Index (CSI)** The CSI, also known as the Threat Score (TS), evaluates the spatial correspondence between the predicted and observed precipitation areas. It is particularly rigorous as it penalizes both false alarms and missed events, making it a standard for structural reliability in nowcasting. The formula is:

$$\text{CSI} = \frac{\text{TP}}{\text{TP} + \text{FP} + \text{FN}}. \tag{15}$$

---

[1] https://tianchi.aliyun.com/dataset/1085

## C. More Experiment Results

### C.1. Detailed quantitative results on the SEVIR dataset

*Table 8.* Detailed quantitative results on the SEVIR dataset across multiple intensity thresholds.

| Models | HSS ↑ | | | | | | CSI ↑ | | | | | |
|---|---|---|---|---|---|---|---|---|---|---|---|---|
| | 16 | 74 | 133 | 160 | 181 | 219 | 16 | 74 | 133 | 160 | 181 | 219 |
| **ConvLSTM** | 0.6567 | 0.6435 | 0.3543 | 0.2173 | 0.1751 | 0.0600 | 0.5770 | 0.5244 | 0.2422 | 0.1402 | 0.1100 | 0.0353 |
| **PredRNN** | 0.6478 | 0.6728 | 0.4273 | 0.2649 | 0.1807 | 0.0699 | 0.6012 | 0.5596 | 0.2982 | 0.1740 | 0.1175 | 0.0444 |
| **PhyDNet** | 0.7698 | 0.7226 | 0.4288 | 0.2860 | 0.2193 | 0.0764 | 0.7054 | 0.6070 | 0.2964 | 0.1871 | 0.1425 | 0.0481 |
| **SimVP** | 0.6990 | 0.6332 | 0.2277 | 0.0374 | 0.0067 | 0.0002 | 0.6314 | 0.5013 | 0.1339 | 0.0194 | 0.0034 | 0.0001 |
| **Earthformer** | 0.7407 | 0.7309 | 0.4358 | 0.2562 | 0.1902 | 0.0859 | 0.6787 | 0.6170 | 0.3014 | 0.1663 | 0.1221 | 0.0526 |
| **MIMO** | 0.7378 | 0.6853 | 0.3501 | 0.2287 | 0.1669 | 0.0433 | 0.6700 | 0.5644 | 0.2279 | 0.1392 | 0.0988 | 0.0235 |
| **DiffCast** | 0.7217 | 0.6524 | 0.3831 | 0.2707 | 0.2271 | 0.1281 | 0.6514 | 0.5325 | 0.2594 | 0.1729 | 0.1421 | 0.0757 |
| **AlphaPre** | 0.7328 | 0.6990 | 0.4034 | 0.2851 | 0.2191 | 0.0921 | 0.6711 | 0.5824 | 0.2771 | 0.1860 | 0.1417 | 0.0575 |
| **Ours** | **0.7872** | **0.7341** | **0.4597** | **0.2935** | **0.2361** | **0.1302** | **0.7176** | **0.6199** | **0.3230** | **0.1972** | **0.1581** | **0.0836** |

Table 8 shows that our method achieves the best performance across all six intensity thresholds under both HSS and CSI metrics. At the low threshold of 16, our model attains an HSS of 0.7872 and CSI of 0.7176, outperforming all baselines. More importantly, at the challenging high threshold of 219, our method maintains an HSS of 0.1302 and CSI of 0.0836, demonstrating superior robustness in capturing extreme precipitation events compared to competing approaches.

### C.2. Inference efficiency and spectral fidelity

Table 9 compares SDIR with representative deterministic and diffusion-based baselines. SDIR achieves the best HSS, CSI, SSIM, and MAE while maintaining substantially lower latency than DiffCast. Although DiffCast obtains the lowest spectral slope error (SSE), it requires iterative denoising and is more than $12\times$ slower than SDIR. In contrast, SDIR provides a favorable accuracy-efficiency trade-off: it significantly improves operational skill scores over Earthformer and DiffCast while preserving strong spectral fidelity and predictable deterministic inference.

*Table 9.* Efficiency and spectral fidelity comparison on the Shanghai dataset. Inference time is measured per sample on a single NVIDIA RTX 4090D GPU. Lower MAE and SSE are better, while higher HSS, CSI, and SSIM are better.

| Model | HSS | CSI | SSIM | MAE | Params | Time | SSE |
|---|---|---|---|---|---|---|---|
| Earthformer | 0.5015 | 0.3711 | 0.7643 | 1395.8 | **1.52M** | **0.16s** | 1.4101 |
| DiffCast | 0.4920 | 0.3628 | 0.8080 | 1450.1 | 49.36M | 14.68s | **0.2302** |
| SDIR (Ours) | **0.5882** | **0.4497** | **0.8548** | **1129.1** | 34.77M | 1.17s | 0.4592 |

### C.3. Performance under extreme precipitation

To evaluate the robustness of SDIR under severe weather conditions, we rank all 4,000 CIKM test sequences by their maximum reflectivity and select the top 9.6% of the sequences, resulting in 384 extreme cases. As shown in Table 10, SDIR consistently outperforms Earthformer and DiffCast on this challenging subset, achieving higher HSS, CSI, and SSIM as well as lower MAE. These results indicate that the proposed spectral refinement framework remains effective for intense precipitation events, where high-frequency convective structures are particularly important for operational warnings.

### C.4. Cross-domain generalization

The experiments cover diverse geographic regions and radar systems, including Shenzhen in CIKM, Shanghai in the Shanghai dataset, and the contiguous United States in SEVIR. To further evaluate cross-domain generalization, we conduct a zero-shot transfer experiment in which models trained only on SEVIR are directly evaluated on the Shanghai dataset. Table 11 shows that SDIR substantially outperforms Earthformer and DiffCast without retraining, demonstrating that the proposed spectral refinement framework generalizes effectively across radar systems, geographic regions, and climate regimes.

*Table 10.* Performance on extreme precipitation events from the CIKM test set. The subset contains the top 9.6% most extreme sequences ranked by maximum reflectivity.

| Model | HSS | CSI | SSIM | MAE |
|---|---|---|---|---|
| Earthformer | 0.5180 | 0.4643 | 0.5635 | 607.95 |
| DiffCast | 0.4992 | 0.4459 | 0.5425 | 614.74 |
| SDIR (Ours) | **0.5648** | **0.5034** | **0.6312** | **547.24** |

*Table 11.* Zero-shot transfer from SEVIR to Shanghai. Models are trained only on SEVIR and directly evaluated on Shanghai without retraining.

| Model | HSS | CSI | SSIM | MAE |
|---|---|---|---|---|
| Earthformer | 0.4878 | 0.3635 | 0.7124 | 1451.7 |
| DiffCast | 0.4500 | 0.3236 | 0.7887 | 1588.0 |
| SDIR (Ours) | **0.5365** | **0.4032** | **0.8545** | **1132.2** |

## C.5. Ablation on Exposure Bias and Low-Frequency Conditioning

To investigate exposure bias in our iterative refinement framework—where training relies on ground-truth low-frequency conditions but inference uses model-generated ones—we implement a straightforward two-stage curriculum to simulate inference during training. For each sample, the model first produces an intermediate prediction $\hat{Y}^{(1)}$ conditioned on the ground-truth low-frequency prior extracted from $Y$. In the second step, it generates $\hat{Y}^{(2)}$ using an updated prior derived solely from $\hat{Y}^{(1)}$, with losses computed on both outputs against ground truth.

*Table 12.* Ablation on exposure bias mitigation strategies on the Shanghai dataset.

| Models | HSS ↑ | | | | | CSI ↑ | | | | | SSIM ↑ | MAE ↓ |
|---|---|---|---|---|---|---|---|---|---|---|---|---|
| | 20 | 30 | 35 | 40 | AVG | 20 | 30 | 35 | 40 | AVG | | |
| **TwoStage** | **0.7072** | **0.6415** | 0.5638 | 0.4211 | 0.5834 | **0.5815** | **0.4973** | 0.4130 | 0.2866 | 0.4446 | 0.8527 | 1142.6 |
| **Ours** | 0.7033 | 0.6379 | **0.5648** | **0.4466** | **0.5882** | 0.5779 | 0.4952 | **0.4167** | **0.3089** | **0.4497** | **0.8548** | **1129.1** |

As shown in Table 12, while the two-stage rollout yields slightly better results at lower intensity thresholds (20 and 30), our model consistently outperforms it at higher, more critical intensities (35 and 40) and achieves superior average metrics. Furthermore, our approach attains higher structural similarity and lower pixel-wise error. This confirms that our standard training strategy is robust against exposure bias, outperforming the rollout-based method in forecasting severe conditions and preserving overall spatial quality.

## C.6. Sensitivity to the PCPSD weight $\eta$

The coefficient $\eta$ balances the magnitude of the PCPSD loss against the spatial reconstruction losses. As shown in Table 13, too small a value under-supervises spectral structure, while too large a value over-penalizes spectral discrepancies and conflicts with spatial regression. The default value $\eta = 0.01$ achieves the best overall performance, confirming that moderate spectral supervision complements spatial accuracy.

*Table 13.* Sensitivity to the PCPSD weight $\eta$ on the Shanghai dataset.

| $\eta$ | HSS | CSI | SSIM | MAE |
|---|---|---|---|---|
| 0.001 | 0.5875 | 0.4458 | 0.8531 | 1214.6 |
| 0.01 | **0.5882** | **0.4497** | **0.8548** | **1129.1** |
| 0.05 | 0.5574 | 0.4193 | 0.8499 | 1161.5 |
| 0.1 | 0.5405 | 0.4016 | 0.8430 | 1338.6 |

## C.7. Additional Qualitative Results

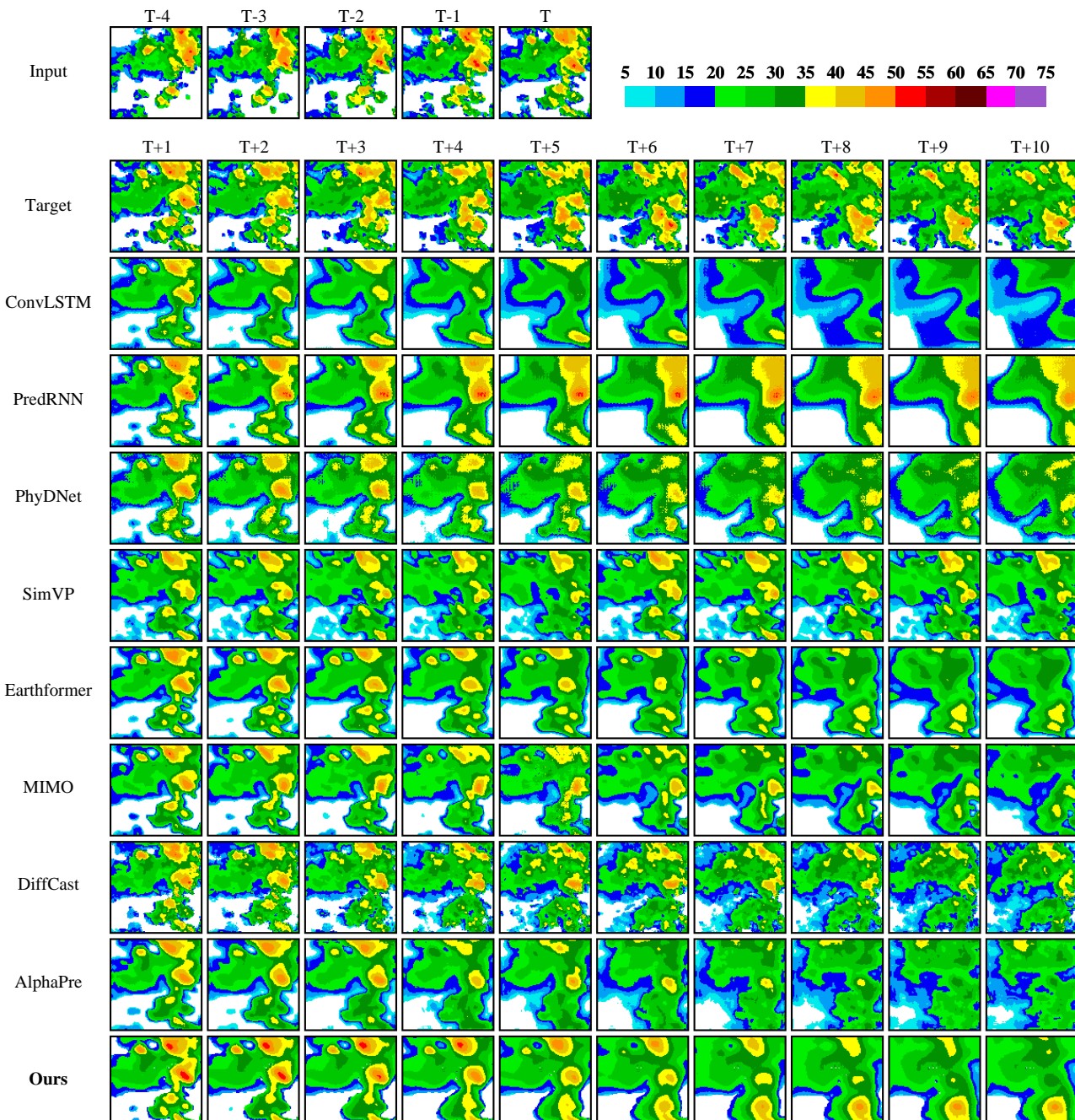

*Figure 5.* Extended qualitative comparison on the CIKM dataset. Regression-based models (e.g., ConvLSTM, SimVP) exhibit progressive blurring and loss of high-reflectivity cores beyond T+6, while DiffCast introduces fragmented artifacts despite recovering some textural detail. Our method consistently preserves the morphology and intensity of convective cells throughout the full prediction horizon, remaining closest to the ground truth target.

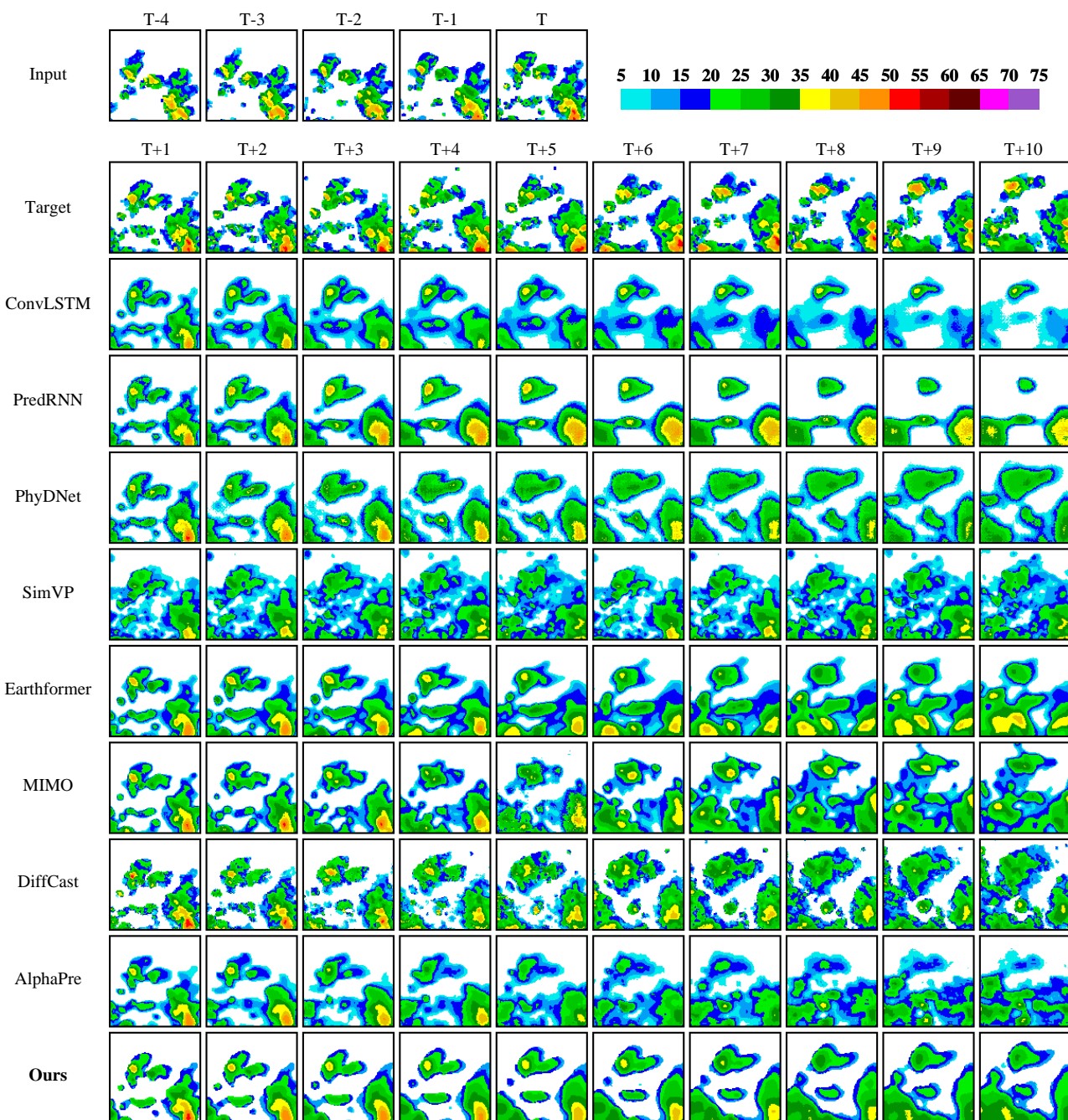

*Figure 6.* Extended qualitative comparison on the CIKM dataset. In this convective development scenario, high-reflectivity cells intensify progressively from T+4 onward. Regression-based models fail to capture this intensification and produce increasingly blurred predictions, while DiffCast generates spatially misaligned structures. Our method better preserves the location and intensity of convective cells across the full prediction horizon.

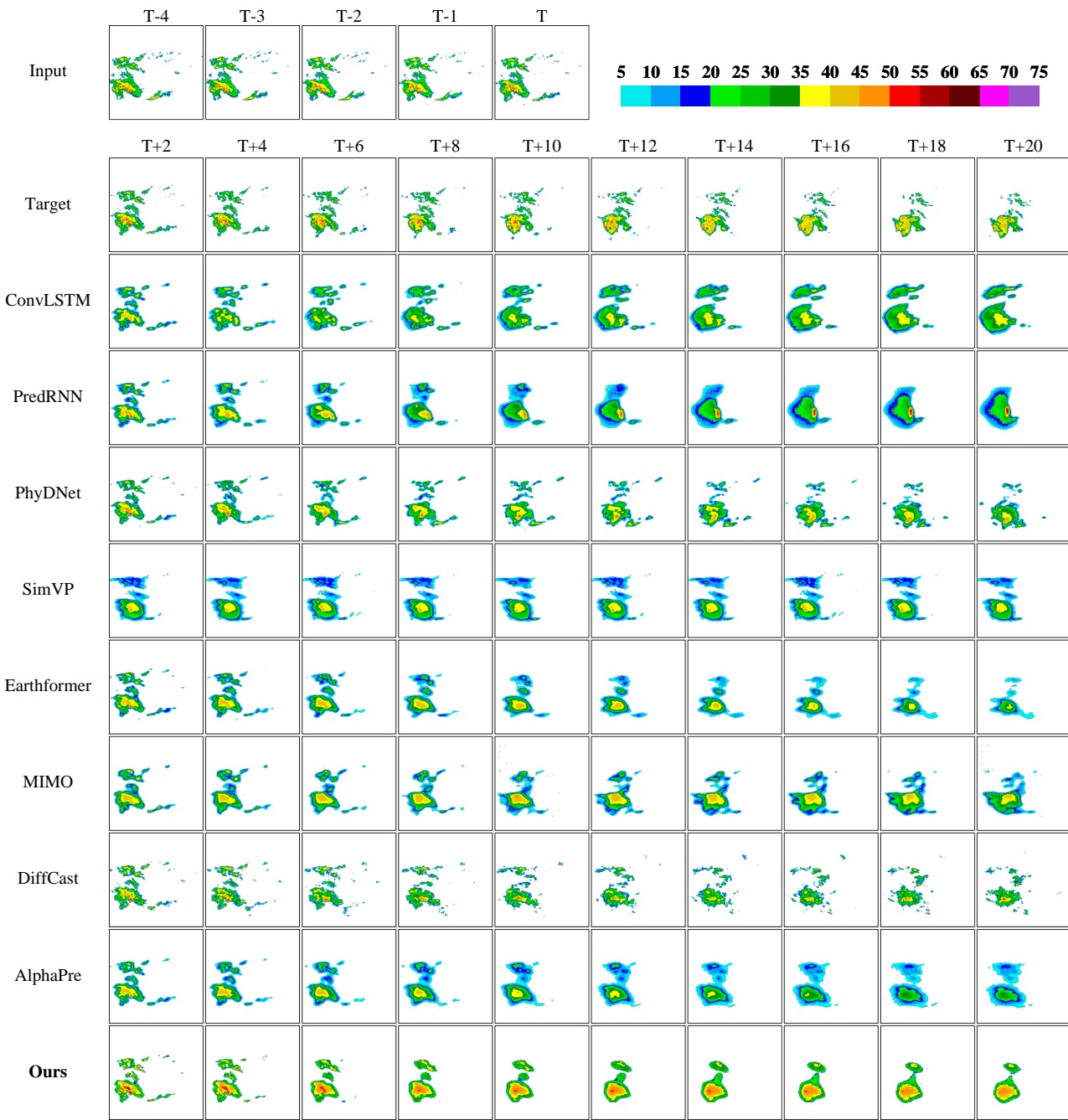

*Figure 7.* Extended qualitative comparison on the Shanghai dataset. This case features multiple scattered convective cells that are inherently challenging to track over extended lead times. Most baseline models either merge distinct cells into blurred blobs (e.g., PredRNN, SimVP) or lose structural coherence entirely beyond T+10 (e.g., DiffCast). Our method better maintains the spatial distribution and intensity of individual convective cells throughout the full prediction horizon.

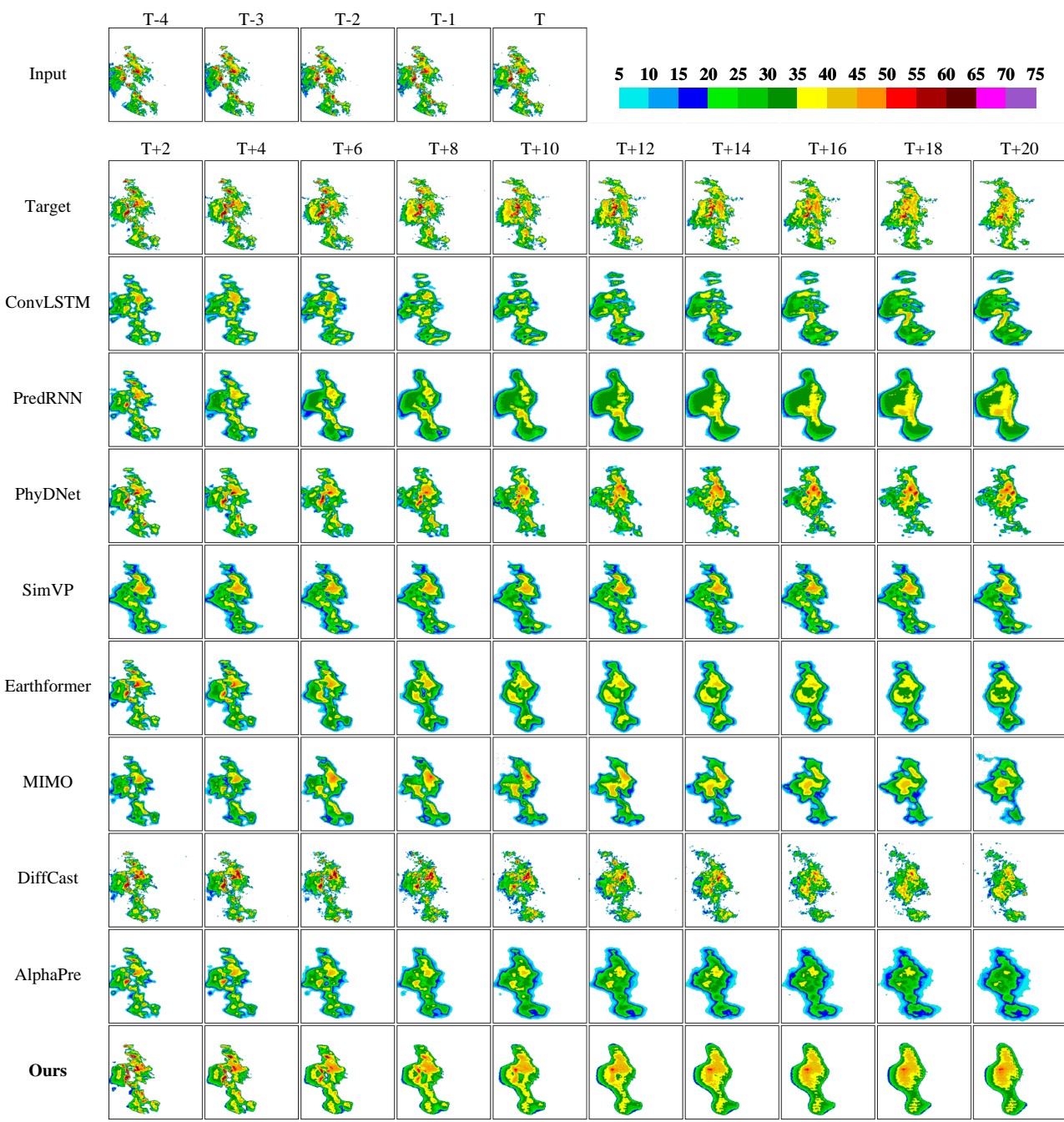

*Figure 8.* Extended qualitative comparison on the Shanghai dataset. The target features a large convective system with a sustained high-reflectivity core. Most baselines progressively smooth out the storm boundaries and lose peak intensity beyond T+8, while AlphaPre exhibits notable structural deformation. Our method consistently preserves the sharp boundaries and high-reflectivity core of the convective system across all lead times.

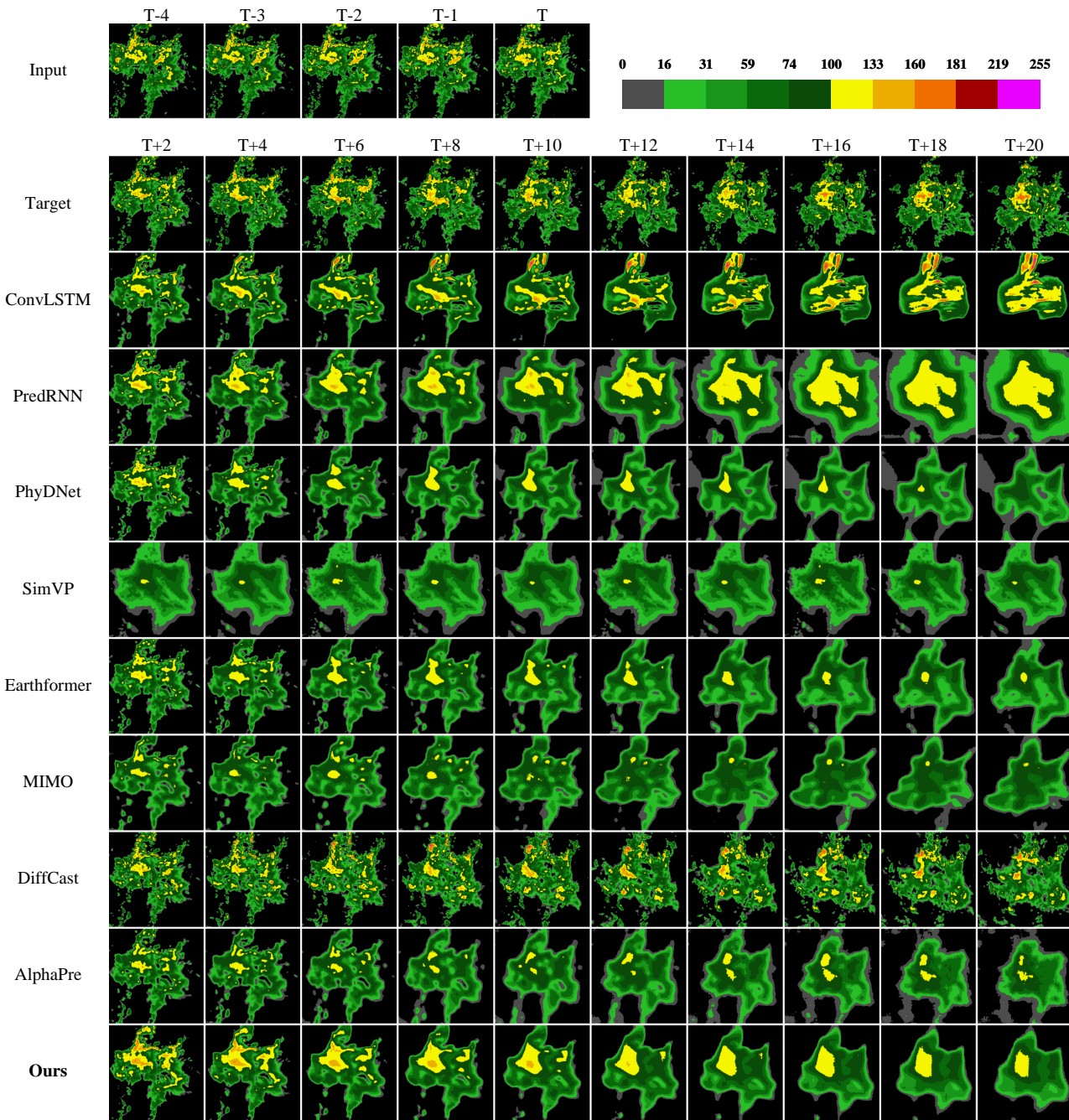

*Figure 9.* Extended qualitative comparison on the SEVIR dataset. The target features a large, complex storm system with sustained high-VIL cores (yellow regions). Most regression-based models (e.g., PhyDNet, SimVP, Earthformer) suffer severe over-smoothing, causing the high-intensity cores to fade into uniform low-reflectivity fields. PredRNN retains some intensity but exhibits substantial boundary expansion and shape distortion. DiffCast recovers finer textures but produces spatially misaligned structures. Our method best preserves the intensity, shape, and internal structure of the convective system across all lead times.

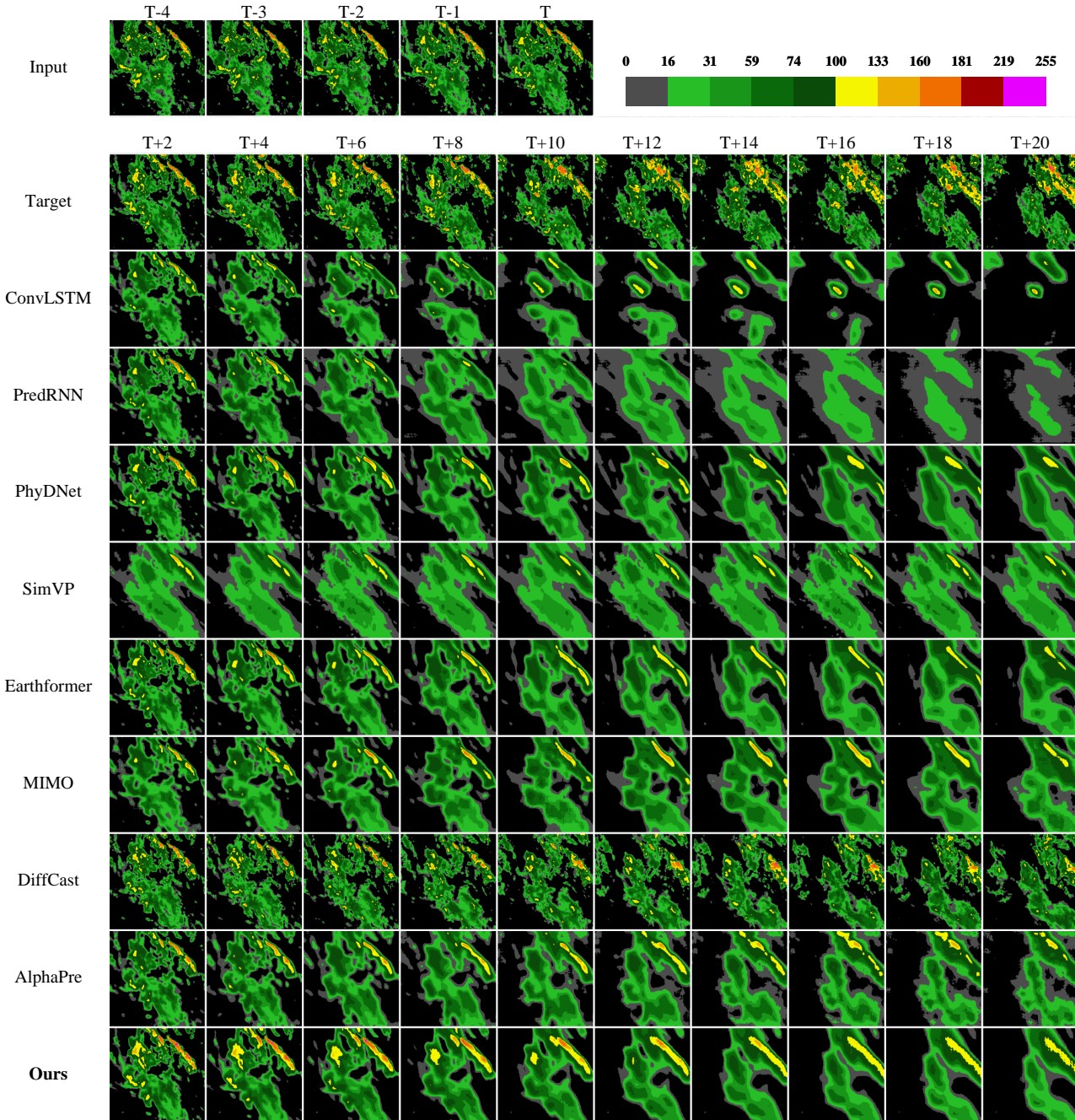

*Figure 10.* Extended qualitative comparison on the SEVIR dataset. The target features a band-shaped high-VIL structure that translates and persists. ConvLSTM fails to maintain the linear morphology beyond T+10, while most regression-based models exhibit severe boundary diffusion that destroys the band structure. DiffCast introduces spurious structures inconsistent with the target's linear orientation. Our method faithfully tracks the translation and intensity of the squall line, preserving its elongated shape and high-VIL cores across all lead times.

