# OpenReview forum: "Learning to Refine: Spectral-Decoupled Iterative Refinement Framework for Precipitation Nowcasting"
_ICML.cc/2026/Conference — ICML 2026 regular_

### Official Review · Reviewer_tWcz · 2026-03-10

**Soundness:** 3
**Presentation:** 3
**Significance:** 4
**Originality:** 3
**Overall Recommendation:** 4
**Confidence:** 3

**Summary:**

The authors propose Spectral-Decoupled Iterative Refinement (SDIR), a deterministic framework that decomposes the forecasting process into progressive spectral refinement stages. The method first extracts a stable low-frequency synoptic skeleton, then iteratively reconstructs high-frequency convective details through a refinement process. The architecture combines two main components: the Synoptic Frequency-Guided Former (SFG-Former) for capturing large-scale spatio-temporal structure and the Fourier Residual Refiner (FR-Refiner) for synthesizing fine spectral details. Additionally, a Physically-Consistent Power Spectral Density (PCPSD) loss is introduced to enforce turbulence-consistent spectral distributions. Experiments on multiple nowcasting benchmarks demonstrate that SDIR improves spatial accuracy while maintaining spectral fidelity comparable to diffusion-based approaches.

**Compliance With Llm Reviewing Policy:**

Affirmed.

**Key Questions For Authors:**

1. Comparison with diffusion-based approaches. How does SDIR compare with recent diffusion-based precipitation forecasting models in terms of both forecasting accuracy and computational efficiency?

2. Sensitivity to refinement steps. How sensitive is the model performance to the number of refinement iterations, and what is the optimal trade-off between accuracy and inference time?

3. Generalization across datasets and climates. Have the authors evaluated the method on datasets from different geographic regions or radar systems to assess generalization across diverse weather patterns?

**Limitations:**

Yes

**Strengths And Weaknesses:**

Strengths

-	Clear motivation. The paper clearly identifies the trade-off between deterministic regression models and generative diffusion models, framing precipitation nowcasting as a spectral refinement problem.

-	Physically motivated modeling design. The proposed framework incorporates spectral decomposition and a power spectral density constraint, providing a principled way to enforce physically consistent predictions.

-	Strong empirical validation on multiple benchmarks. Experiments show consistent improvements in key metrics such as HSS, CSI, and SSIM, demonstrating that iterative refinement can effectively recover high-frequency convective structures.

Weaknesses

-	Computational overhead from iterative refinement. The proposed framework requires multiple refinement steps, which may increase inference latency compared with one-shot prediction models.

-	Limited comparison with recent generative nowcasting methods. The evaluation could be strengthened by including stronger generative baselines such as diffusion-based weather forecasting models.

-	Dependence on spectral loss design. The PCPSD loss plays a crucial role in the method, but the sensitivity of performance to its design choices and hyperparameters is not extensively analyzed.

-	Limited discussion of operational deployment. While the method improves spectral fidelity and spatial accuracy, the paper provides limited analysis of its applicability in real-time operational meteorological systems.

---

> ### Author Rebuttal · Authors · 2026-03-31
>
> We thank the reviewer for the valuable and constructive comments.
>
> **Q1 & W2: Comparison with diffusion-based models**
>
> In our experiments, we include DiffCast, a SOTA diffusion-based residual precipitation nowcasting model, as a representative generative baseline on all three benchmarks. As summarized in the main paper and in Table 1 provided in our response to Reviewer ffKk, SDIR consistently outperforms DiffCast and Earthformer across all benchmarks. At the same time, SDIR offers a substantially more favorable efficiency profile: its deterministic multi-step refinement is significantly faster than DiffCast’s iterative denoising, which leads to lower and more predictable latency in operational settings.
>
> **Q2 & W1: Computational overhead and sensitivity to refinement steps**
>
> We thank the reviewer for these practical concerns and address both with Table 5 below.
>
> **Table 5: Performance and Inference Time across Refinement Steps**
> |Steps|HSS|CSI|SSIM|MAE|Inference time|
> |-|-|-|-|-|-|
> |1-step|0.5584|0.4243|0.8522|1111.0|0.30s|
> |8-step (ours)|0.5882|0.4497|0.8548|1129.1|1.17s|
> |16-step|0.5898|0.4487|0.8533|1190.1|2.14s|
> |32-step|0.5564|0.4164|0.8475|1352.6|4.09s|
>
> Step sensitivity. Performance improves steadily from 1- to 8-step, then degrades beyond that due to over-refinement. We select 8-step as our default, which achieves the best balance across all metrics.
>
> Computational overhead. We highlight three practical points:
>
> * The 1-step variant is directly comparable to one-shot models and already outperforms most baselines.
>
> * Our 8-step default (1.17s) offers strong categorical accuracy with low inference cost, well-suited for operational deployment.
>
> * All configurations run well within the 6-minute radar update interval, imposing no operational bottleneck. Compared to DiffCast (14.68s per sequence), SDIR's overhead is substantially lower.
>
> **Q3: Generalization across datasets and climates**
>
> Our experiments already cover diverse geographic regions and radar systems, including:
> * CIKM: Shenzhen, China
> * Shanghai: Shanghai, China
> * SEVIR: Contiguous United States
>
> Across all three benchmarks, SDIR achieves consistent SOTA performance, demonstrating strong robustness to varying weather patterns and data distributions.
>
> To further evaluate cross-domain generalization, we conduct a zero-shot transfer experiment (Table 6), where a model trained solely on SEVIR is directly evaluated on the Shanghai dataset.
>
> **Table 6: Zero-Shot Transfer Performance (SEVIR to Shanghai)**
> |Model|HSS|CSI|SSIM|MAE|
> |-|-|-|-|-|
> |Earthformer|0.4878|0.3635|0.7124|1451.7|
> |DiffCast|0.4500|0.3236|0.7887|1588.0|
> |SDIR (ours)|0.5365|0.4032|0.8545|1132.2|
>
> SDIR significantly outperforms both Earthformer and DiffCast in this setting, demonstrating that the proposed spectral refinement framework generalizes effectively across radar systems, geographic regions, and climate regimes without retraining.
>
> **W3: Sensitivity of PCPSD Loss to Design Choices and Hyperparameters**
>
> We analyze the robustness of the PCPSD loss from both design and empirical perspectives.
>
> 1. Physically motivated design (not arbitrary).
>
>     * Log-spectral domain: emphasizes relative energy differences across scales; linear supervision would be dominated by low-frequency energy.
>     * Radial averaging: aligns with the approximate isotropy of mesoscale precipitation and provides a noise-robust spectral representation.
>     * Dynamic masking: ensures consistency with the frequency-unlocking curriculum, preventing conflicting gradients from prematurely supervising high-frequency bands.
>
>     These components are derived from physical principles of turbulence and spectral energy cascades, rather than heuristic choices.
>
> 2. Hyperparameter Sensitivity.
> We have conducted additional sensitivity experiments on the two most critical hyperparameters: the base coefficient $\eta$ and the curriculum sampling parameter $\beta$. Please refer to our response to Reviewer ffKk.
>
> **W4 & Limitations: Operational deployment**
>
> We address the key operational requirements as follows:
> * Inference latency. As shown in Table 5, SDIR completes inference in 1.17s, which is well within the typical 5–6 minute radar update interval of operational nowcasting systems.
> * Deterministic output. Unlike diffusion-based methods that require ensemble sampling to produce reliable forecasts, SDIR is fully deterministic, generating a single physically-consistent prediction per forward pass.
> * Hardware accessibility. All experiments are conducted on 4× RTX 4090 D GPUs, which represent commodity hardware increasingly available in operational meteorological centers, rather than large-scale HPC clusters.
> * System compatibility. These properties make SDIR directly compatible with existing radar-based nowcasting pipelines, where predictable latency, deterministic outputs, and stable runtime are critical for real-time deployment.

---

> > ### Author Rebuttal · Reviewer_tWcz · 2026-04-03
> >
> > The rebuttal has addressed my concerns, so I maintain my initial rating (weak accept).

---

> > > ### Author Response · Authors · 2026-04-04
> > >
> > > We sincerely thank the reviewer for reviewing our rebuttal and confirming that the concerns have been fully resolved. We deeply appreciate your time, the constructive feedback provided throughout the review process, and your support for our work.

---

### Official Review · Reviewer_ffKk · 2026-03-10

**Soundness:** 3
**Presentation:** 3
**Significance:** 3
**Originality:** 3
**Overall Recommendation:** 4
**Confidence:** 4

**Summary:**

This paper proposes SDIR (Spectral-Decoupled Iterative Refinement), a deterministic framework for precipitation nowcasting designed to bridge the gap between regression-based over-smoothing and diffusion-based hallucinations. The basic idea is to decompose the forecasting task into a coarse-to-fine spectral refinement process. The framework uses a dual-path strategy. A DCT-truncated conditioning is used to extract the low-frequency synoptic skeleton, while an FR-Refiner with Scale-Conditioned Fourier Neural Operators (SFNO) progressively corrects high-frequency residual textures. To ensure physical consistency, a PCPSD loss with dynamic masking is used to enforce turbulence-consistent spectral distributions. The method shows consistent improvements in HSS, CSI, SSIM, and MAE over eight baselines across CIKM, Shanghai, and SEVIR benchmarks.

**Compliance With Llm Reviewing Policy:**

Affirmed.

**Final Justification:**

SDIR presents a well motivated spectral refinement framework for precipitation nowcasting, grounding the blurring problem in spectral energy collapse rather than pixel level degradation. The physical intuition is clear and the design choices follow from it naturally.

The rebuttal comprehensively addressed my original concerns. The new experiments on inference efficiency, tail performance, and hyperparameter sensitivity add meaningful empirical support, and the SSIM versus HSS and CSI discrepancy is convincingly explained. The SFNO architectural details are now sufficiently clarified for reproducibility.

While the novelty lies in the physically motivated combination of existing components rather than a single new building block, the execution is careful and the practical benefits are real.

Overall, the rebuttal has strengthened my confidence in the work and I support acceptance.

**Key Questions For Authors:**

1. Can you please provide inference latency and parameter count comparisons against DiffCast and Earthformer? Without runtime data, it is difficult to assess the trade-off for operational deployment.
2. The PCPSD loss boosts HSS/CSI significantly but improves SSIM only marginally. Is the spectral loss primarily improving categorical detection skill, or is SSIM simply insensitive to the targeted spectral improvements?
3. Have you evaluated SDIR specifically on the top percentile of precipitation extremes or rapid intensification events to fully characterize tail performance?
4. How sensitive is the performance to the Beta parameterization? Was this chosen via grid search, and what is the degradation under different configurations?

**Limitations:**

The authors acknowledge lower fine-scale texture realism compared to generative models quite well. However, there are some additional gaps. Being an operational framework, I think the lack of computational cost reporting is a significant omission. Furthermore, the spectral fidelity claims lack quantitative support in the main evaluation. The framework produces deterministic point forecasts with no uncertainty quantification, which limits its utility for risk-based decision-making. Finally, the current evaluation does not include a tail-conditioned metric subset, leaving the model's behavior on the highest-stakes events incompletely characterized.

**Strengths And Weaknesses:**

Soundness:
The physical motivation used to develop the framework is quite interesting. The energy cascade analogy—stabilizing synoptic scales before refining convective textures is theoretically well-justified and directly informs the DCT curriculum and PCPSD loss. The main thing I find interesting is that the framework connects the spectral energy decay to physical reality rather than relying on purely engineering-driven methods. However, there are a few concerns in the reported results:
1. I think addition of quantitative metrics like spectral slope error or PSD divergence to the main tables would be interesting. If spectral quality is a primary contribution, it can be a dedicated evaluation column.
2. The PCPSD loss introduces a hyperparameter \eta, but its value is not stated and no sensitivity analysis is provided. Similarly, you did not show any experiments for different Beta setting.
3. The description of the SFNO block of the FR-Refiner omits key details, such as the number of retained Fourier modes and whether the SoftShrink threshold is learned or fixed.

Presentation:
The paper is well-written and the figures are descriptive and informative. I liked Figure 2, which illustrates the dual-path architecture overview quite well, and lead-time decay plots are very compelling. However, there are a few gaps. The loss weight \eta is never defined in the main text, and there is an occasional notation inconsistency regarding the zero initial condition in Figure 2(d) versus the text  (algo 2: C_s =0). Additionally, the lack of direct comparison with the broader FNO or neural operator literature feels like a missed opportunity given the centrality of SFNO to the design.

Significance:
The paper addresses a genuine and practically important problem. The framing of the blurring problem as a spectral energy collapse rather than a pixel-level is a conceptually interesting contribution. The consistent performance margins at the 40 dBZ threshold have direct operational relevance for extreme precipitation alerting. The deterministic design also gives SDIR a practical advantage over diffusion-based alternatives in latency-sensitive settings, though this computational advantage is never quantified.

Originality:
The authors presents a novel approach that combines DCT-based spectral curriculum, Scale-Adaptive Transformers, and SFNO-based residual synthesis. The key insight is that the degradation in forecasting quality is a problem of maintaining spectral energy across scales rather than just a lack of model capacity. This perspective differentiates the work from standard sharpening methods. While individual components like SFNO, AdaLN, and RoPE are drawn from existing work, the novelty lies primarily in their physically motivated combination and the frequency-unlocking inference scheme.

---

> ### Author Rebuttal · Authors · 2026-03-31
>
> We sincerely thank the reviewer for the detailed and insightful comments, which will substantially improve the quality of the paper.
>
> **Q1 & W1: Inference efficiency and quantitative spectral metrics**
>
> We agree that both operational efficiency and quantitative spectral evaluation are critical. We report accuracy, parameter counts, per-sample inference time, and spectral slope error (SSE) on the Shanghai dataset using a single NVIDIA RTX 4090 D GPU (see Table 1).
>
> **Table 1: Efficiency and spectral fidelity comparison**
> |Model|HSS|CSI|SSIM|MAE|Params|Inference time|SSE|
> |-|-|-|-|-|-|-|-|
> |Earthformer|0.5015|0.3711|0.7643|1395.8|1.52M|0.16s|1.4101|
> |SDIR (ours)|0.5882|0.4497|0.8548|1129.1|34.77M|1.17s|0.4592|
> |DiffCast|0.4920|0.3628|0.8080|1450.1|49.36M|14.68s|0.2302|
>
> SDIR offers a highly favorable accuracy-efficiency trade-off: it achieves superior skill scores and is over 12× faster than DiffCast. Moreover, SDIR matches DiffCast's high spectral fidelity, significantly outperforming Earthformer. Due to the short rebuttal period, we have computed the SSE specifically for SDIR, DiffCast, and Earthformer.
>
> **Q2: PCPSD boosts HSS/CSI significantly but SSIM only marginally**
>
> The discrepancy arises because SSIM measures structural similarity in pixel space, which is insensitive to spectral energy redistribution. HSS and CSI, being threshold-based categorical metrics, are directly sensitive to recovered high-frequency energy around convective cores, which raises true positives at high-reflectivity thresholds (30–40 dBZ). This confirms that PCPSD targets precisely the spectral deficiency that matters for operational warning — detection of intense precipitation — rather than broad pixel-level changes.
>
> **Q3: Tail performance on precipitation extremes**
>
> Following the reviewer's suggestion, we conducted a tail performance analysis on the CIKM test set. We ranked all 4,000 test sequences by their maximum reflectivity and selected the top 9.6% most extreme sequences (384 sequences) as a held-out extreme subset. Table 2 compares SDIR, DiffCast, and Earthformer on this subset, demonstrating that SDIR maintains its performance advantage under extreme precipitation conditions.
>
> **Table 2: Performance on extreme precipitation events**
> |Model|HSS|CSI|SSIM|MAE|
> |-|-|-|-|-|
> |Earthformer|0.5180|0.4643|0.5635|607.95|
> |DiffCast|0.4992|0.4459|0.5425|614.74|
> |SDIR (ours)|0.5648|0.5034|0.6312|547.24|
>
> **Q4 & W2: Hyperparameter η and β sensitivity**
>
> We will explicitly state the default values $\eta = 0.01, \alpha = 1.0, \beta = 3.0$ in Section 4.3 and add a “Hyperparameter Sensitivity Analysis” subsection in Appendix D.
>
> The base coefficient $\eta$ in $\phi(s) = \eta (s/W)^2$ is used purely to balance the magnitude of the PCPSD loss against the spatial L1 loss. We have now included a sensitivity analysis for $\eta$ on the Shanghai dataset (Table 3). Smaller $\eta$ under-supervises spectral structure, while larger $\eta$ over-penalizes and conflicts with spatial regression. Our default $\eta = 0.01$ strikes the best balance, confirming that moderate spectral supervision complements rather than conflicts with spatial accuracy.
>
> **Table 3: Sensitivity to the PCPSD weight $\eta$**
> |$\eta$|HSS|CSI|SSIM|MAE|
> |-|-|-|-|-|
> |0.001|0.5875|0.4458|0.8531|1214.6|
> |0.01 (ours)|0.5882|0.4497|0.8548|1129.1|
> |0.05| 0.5574|0.4193|0.8499|1161.5|
> |0.1| 0.5405|0.4016|0.8430|1338.6|
>
> Sensitivity to $\text{Beta}$ distribution β (Table 4). Our chosen $\text{Beta}(1.0, 3.0)$ achieves the best HSS and CSI. This confirms that a moderate low-frequency prioritization is theoretically necessary: the synoptic skeleton must be stabilized before high-frequency residuals can be coherently learned.
>
> **Table 4: Sensitivity to the Beta distribution $\beta$**
> |Distribution|HSS|CSI|SSIM|MAE|
> |-|-|-|-|-|
> |Beta(0.8, 1.5)|0.5606|0.4206|0.8485 |1299.2|
> |Beta(1.0, 2.5)|0.4835|0.3759|0.8493|1241.5|
> |Beta(1.0, 3.0) (ours)|0.5882|0.4497|0.8548|1129.1|
> |Beta(1.0, 3.5)|0.5582|0.4222|0.8543|1140.5|
>
> Overall, both hyperparameters exhibit clear and interpretable optima, demonstrating that SDIR's design choices are principled and robust rather than arbitrarily tuned.
>
> **W3: SFNO block details**
>
> Thank you for requesting this clarification. The FR-Refiner employs 8 SFNO blocks at its latent bottleneck. Following the architectural design of FourCastNet, the SoftShrink threshold is a fixed scalar set to 0.01 (not learned). To ensure reproducibility, the complete SDIR source code, pre-trained weights, and exact configuration details will be publicly released upon acceptance.
>
> We will carefully revise the manuscript according to these suggestions, including rechecking the entire paper for inconsistencies and errors (such as the notation in Figure 2(d)) and adding a broader FNO literature introduction. All such issues will be corrected in the camera-ready version.

---

> > ### Author Rebuttal · Reviewer_ffKk · 2026-04-02
> >
> > Thank you for the detailed rebuttal. The additional experiments on inference latency (Table 1), tail performance (Table 2), and hyperparameter sensitivity (Tables 3 & 4) directly address the core concerns raised and significantly strengthen the paper. The explanation for the SSIM vs HSS/CSI discrepancy under the PCPSD loss is also technically convincing.
> >
> > One remaining concern is that the rebuttal still does not specify the number of retained Fourier modes in the SFNO block, which is important for reproducibility. While the number of blocks and SoftShrink threshold are clarified, this detail should be explicitly included in the revised manuscript.
> >
> > The proposed revisions (notation fixes, sensitivity appendix, and expanded FNO literature discussion) are appropriate. Overall, the rebuttal substantially improves the paper, and the remaining issues are limited and can be addressed in revision.

---

> > > ### Author Response · Authors · 2026-04-02
> > >
> > > Thank you for your acknowledgement. We are pleased that the additional experiments and clarifications have proven helpful in addressing your concerns.
> > >
> > > With respect to the retained Fourier modes, we sincerely apologize for the repeated omission in our earlier description. In our implementation, the SFNO block does not apply hard frequency truncation, meaning it retains **all Fourier modes**. Because we utilize a 2D real-to-complex FFT on the spatial dimensions of the bottleneck, the full spectrum is preserved along the height axis, while the width axis leverages Hermitian symmetry. Therefore, the exact number of complex frequency coefficients preserved is $H_b \times (\lfloor W_b / 2 \rfloor + 1)$. This results in 32 × 17 modes for the CIKM dataset (bottleneck resolution 32×32) and 64 × 33 modes for Shanghai/SEVIR (bottleneck resolution 64×64). We will explicitly include this information in Section 4.3 of the revised manuscript.
> > >
> > > Thank you once again for your positive feedback. We sincerely hope the clarifications above will further strengthen your confidence in the quality of our work.

---

### Official Review · Reviewer_SEP4 · 2026-03-12

**Soundness:** 3
**Presentation:** 3
**Significance:** 2
**Originality:** 3
**Overall Recommendation:** 4
**Confidence:** 3

**Summary:**

This paper proposes a Spectral-Decoupled Iterative Refinement method for precipitation nowcasting. Specifically, SDIR employs a dual-path framework: the Synoptic Frequency-Guided Former and the Fourier Residual Refiner. In addition, a Physically-Consistent Power Spectral Density loss with dynamic masking enforces turbulence-consistent spectral distribution.

**Compliance With Llm Reviewing Policy:**

Affirmed.

**Final Justification:**

After carefully reading the paper and the author's response, I think that the overall score of this paper should be 4.

**Key Questions For Authors:**

1. The relationships among the different paradigms illustrated in Fig. 1 are somewhat confusing, making it difficult to intuitively understand the conceptual differences and connections between them.
2. The frequency-decoupled refinement lacks rigorous theoretical analysis explaining.
3. Some theories such as PCPSD loss require detailed experiments and analysis to prove.
4. The relationship between the training procedure and the inference pipeline in the proposed framework is not sufficiently clear and should be further clarified.
5. The rationality and applicability of PCPSD loss should be analyzed.

**Limitations:**

Yes

**Strengths And Weaknesses:**

Strengths:
1. The paper clearly identifies the fundamental trade-off between regression-based methods and diffusion-based approaches. The proposed framework is a well-motivated perspective for precipitation nowcasting.
2. The proposed dual-path architecture combines global and spectral refinement.

Weaknesses:
1. The relationships among the different paradigms illustrated in Fig. 1 are somewhat confusing, making it difficult to intuitively understand the conceptual differences and connections between them.
2. The frequency-decoupled refinement lacks rigorous theoretical analysis explaining.
3. Some theories such as PCPSD loss require detailed experiments and analysis to prove.
4. The relationship between the training procedure and the inference pipeline in the proposed framework is not sufficiently clear and should be further clarified.
5. The rationality and applicability of PCPSD loss should be analyzed.

---

> ### Author Rebuttal · Authors · 2026-03-31
>
> We sincerely thank the reviewer for this constructive feedback.
>
> **Q1: Clarity of Fig. 1 — relationships among paradigms**
>
> We agree and have completely redesigned Figure 1 with the following improvements:
>
> 1. Restructured Layout: Replaced the centralized layout with a parallel top-to-bottom matrix, presenting all three paradigms in a standardized "Input → Process → Output" pipeline for direct comparison.
> 2. Clarified Mechanisms: Visually highlighted the distinct processing mechanisms of each paradigm to make conceptual differences immediately apparent.
> 3. Contextual Evidence: Relocated the PSD plot into section (a) to directly support the "Spectral Decay" claim, and placed the GT into section (b) as an immediate visual contrast against diffusion hallucinations.
>
> The revised Figure 1 is available at: https://anonymous.4open.science/r/SDIR-Figure1/figure1.pdf
>
> **Q2: Rigorous theoretical analysis of frequency-decoupled refinement**
>
> We will expand our analysis to explicitly link our theory to these empirical proofs:
> 1. Physical Grounding: Inspired by the atmospheric energy cascade ($S(k) \propto k^{-5/3}$), our design mirrors kinetic energy transfer from synoptic systems to convective eddies. Stabilizing the low-frequency skeleton $C_s = \text{IDCT}(\text{Trunc}_s(\text{DCT}(Y)))$ before injecting high-frequency textures ensures structural consistency and prevents fine-scale structures from violating the large-scale flow.
> 2. Optimization: Pixel-wise losses inherently bias models toward low-frequency means, causing spectral decay. Our objective $L = L_\text{base} + L_\text{res} + \phi(s) L_\text{pcpsd}$ alleviates cross-scale gradient competition. The base path focuses on global modes, while the residual path recovers high-frequency details, guided by a dynamic spectral penalty $\phi(s) = \eta \cdot (s/W)^2$.
> 3. Inference Dynamics: The frequency-unlocking schedule $\{s_1, \ldots, s_K\}$ acts as a deterministic scale-space continuation. Rather than an ill-posed single-step prediction, the model iteratively tracks $\hat{Y}^{(k)} = \text{SDIR}(X, \hat{Y}^{(k-1)}, s_k)$, constraining each step to a well-posed subproblem. As demonstrated by our results, this coarse-to-fine conditioning ensures stable evolution and prevents meteorological hallucinations.
>
> **Q3: Theoretical and experimental support for PCPSD loss**
>
> We agree that the theoretical claims regarding the PCPSD loss require empirical validation. We address this through our existing ablations and newly added sensitivity experiments:
>
> 1. Ablation Results: Integrating PCPSD boosts HSS by 9.6% and CSI by 10.8%. The slight, expected MAE increase confirms our theory: pixel-wise metrics inherently penalize the sharp, high-frequency structures that PCPSD explicitly restores.
> 2. Spectral Validation: Our 1D radially-averaged PSD analysis (Figure 1) proves that PCPSD mathematically aligns the model's energy distribution with the ground truth spectrum, effectively mitigating spectral decay.
> 3. Sensitivity Analysis: New evaluations of the base coefficient ($\eta$) and curriculum sampling ($\beta$) (see Reviewer ffKk response) empirically validate our mechanics. $\eta$ controls the spatial-spectral trade-off, while $\beta$ proves our coarse-to-fine curriculum is a theoretically necessary stabilizer, not a mere heuristic.
>
> **Q4: Relationship between training procedure and inference pipeline**
>
> We will add a dedicated clarification paragraph. The key distinction is:
>
> * Training (Algorithm 1): The low-frequency condition $C_s$ is derived *from the ground truth* $Y$ via DCT truncation at scale $s$ sampled from $\text{Beta}(1.0, 3.0)$. This provides clean oracle supervision so each component learns its designated frequency range reliably.
> * Inference (Algorithm 2): Starting from a *cold-start* condition $C_s = 0$, the model iteratively increments the frequency scale through a predefined schedule $S = \{s_1, s_2, \dots, s_K\}$. At each step $k$, the current prediction $\hat{Y}^{(k)}$ is re-truncated at scale $s_{k+1}$ to generate the condition for the next iteration—replacing the ground-truth oracle with the model's own output.
>
> **Q5: Rationality and applicability of PCPSD loss**
>
> We will clarify the rationality and practical applicability of the PCPSD loss in the revised manuscript.
> 1. Rationality: Pixel-wise losses like MSE average out spatial uncertainty, inherently causing over-smoothing. PCPSD resolves this by matching the radially-averaged PSD, forcing the model's energy distribution to obey atmospheric turbulence laws and prevent spectral decay.
> 2. Applicability: PCPSD provides physically principled supervision. Crucially, its dynamic masking aligns perfectly with our curriculum by heavily penalizing deviations in newly unlocked high frequencies while preserving the already-stabilized low-frequency synoptic skeleton. Its robust applicability is validated across three diverse benchmarks.

---

> > ### Author Rebuttal · Reviewer_SEP4 · 2026-04-03
> >
> > Thank you for the detailed rebuttal. The authors have addressed all my concerns. I'd like to maintain my original score 4.

---

> > > ### Author Response · Authors · 2026-04-03
> > >
> > > Thank you for carefully reading our rebuttal and for your constructive feedback throughout the review process. We are glad that our responses addressed your concerns, and we appreciate your continued support of the paper.

---

### Decision · Program_Chairs · 2026-04-30

**Decision:**

Accept (regular)

**Comment:**

The three reviewers supported acceptance with weak accept ratings. After the rebuttal, two reviewers stated that their concerns had been addressed, while the third said the rebuttal substantially strengthened the paper and left only a limited reproducibility issue. Across the reviews, the main concerns were limited theoretical support for the spectral refinement formulation, the need for more direct quantitative validation of the spectral claims, and initially missing details on efficiency, hyperparameter sensitivity, and implementation. At the same time, reviewers generally agreed that the paper is clearly motivated and that the empirical results are strong across multiple benchmarks. On balance, I view the paper as acceptable, although the case rests more on the method and experimental evidence than on a fully established theoretical contribution.